# Electrical appliances moderate households' water demand response to heat

Alberto Salvo[1]

Analysis of potentially interconnected residential water and energy demand is sparse. In a 1-in-10 random sample of Singapore households living in apartments, water use per capita declines over the socioeconomic distribution, whereas electricity use rises. Here I show that in this leading Asian city and tropical climate, water and electricity demand respond differentially to heat across different socioeconomic groups. When temperatures rise, water demand increases among lower-income households but remains unchanged among higher-income households. In sharp contrast, heat induces larger shifts in electricity demand among higher-income households. With air-conditioner penetration ranging from 14 to 99% across different socioeconomic groups, my interpretation is that water provides heat relief for households that have yet to adopt air conditioning. How Singaporeans' resource demands respond to heat at different income levels can inform the future responses of a vast urban population on rising incomes living in the water-stressed tropics, in similar and warming climates.

[1] Department of Economics, National University of Singapore, 10 Kent Ridge Crescent, Singapore 119260, Singapore. Correspondence and requests for materials should be addressed to A.S. (email: albertosalvo@nus.edu.sg)

Natural population growth, urbanization, and rising economic activity are expanding water demand in cities around the world[1-4]. At the same time, higher global temperatures and shifting precipitation patterns due to climate change are expected to impact local water supplies, with many densely populated regions in Asia, the Mediterranean, and elsewhere experiencing reduced water availability over extended periods[5-7]. Access to clean water is a key determinant of public health, economic growth, welfare, and regional peace[8-11]. On the demand side, local governments have responded to water stress with both price-based and nonprice-based approaches to water conservation[12,13]. Climate change may impact both the quantity of resources—for example, air conditioning demand—and the quality of resources, such as the reliability of supply, including electricity[14-18].

An established literature examines the determinants of residential water demand, including price, income, and, to a lesser extent, variation in weather, such as rainfall and temperature[19-22]. Likely due to data access, this literature has mostly considered rich-country households, many of which have gardens that require irrigation[23]. Water demand for irrigation is sensitive to weather, and may be difficult to subtract from jointly billed indoor demand. The research design is often cross-sectional, in which households and dwellings may differ in unobserved ways, or based on time series aggregated up to city level. Research has not considered how a household's electricity use from the adoption of appliances such as air conditioners (AC) may modify its weather-induced water demand. Previous researchers of residential water demand have focused their data collection efforts on residential water quantities and prices, and abstracted away from electricity use.

Here, I examine several years' worth of microdata on water and electricity bills for a 1-in-10 random sample of households in Singapore—namely, 120,000 households with a modal 19 bimonthly periods of observation per household (microdata for short). Singapore (population 5.4 million) is a leading, newly affluent island city–state covering 720 km² off Peninsular Malaysia in tropical Asia. It is densely urban, with 94% of households residing in apartments without private yards, allowing me to control (by design) for garden use and instead focus on indoor water use, such as bathing and laundry. Unlike developing countries such as Bangladesh and India, Singapore enjoys modern infrastructure. Residential connections to water and electricity grids are near universal, with single-household meters observable by the billing agent, from whom I have obtained the household-level panel. After decades of prosperity, Singapore displays a wide income distribution, but is essentially free of absolute poverty attributes such as hunger and homelessness. Residents of one- or 2-room apartments, for example, have a mean annual household income per person of US$ 9300, and AC adoption is only 14%. At the other end of the socioeconomic distribution, condominium apartments—which many Singaporeans aspire to—are characterized by a mean household income per person of US$ 68,900, and AC penetration is 99%. The climate is tropical, with daily mean temperatures varying in the 23.7–30.5 °C (74.7–86.9 °F) range, and relative humidity hovering about 80%. This allows me to focus on a range of weather variation to which household water and electricity demands respond, and that is relevant to a large and rising urban middle class[24] living in the tropics in similar and warming climates. Currently, only 8% of the world's tropical population, 3 billion strong, has AC[25].

A recent review calls for more research on adaptive behaviors to global warming using large-scale real-world data[26]. Here, I combine water and electricity (indirectly, AC) use within households over the socioeconomic distribution. Household-specific income, wealth or expenditure are not directly observable in the microdata. Thus, when estimating a household's resource demand response to ambient temperature, I take the household's dwelling type—which I do observe—as a proxy measure of its socioeconomic standing, noting that housing in Singapore is a key component of household wealth. It is important to appreciate that the design controls for all unobserved individual household (and dwelling) characteristics, including time-averaged income, besides seasonality and other environmental factors. I find that hot weather induces individuals to wash more, and that electricity-hungry appliances such as AC can negate this heat-stress-aggravates-water-stress channel. The inferred user behavior is actual, based on the individual choices of 120,000 households over time, and not stated or hypothetical. The response is representative of a leading Asian city that is generally warm, and thus estimates likely inform on long-term adaptation (at current technologies and prices) rather than a one-off response to a heat wave.

## Results

**Analysis of institutions and data from different sources.** Annual expenditure across Singapore's 1.2 million households ranges from less than US$ 10,000 to over US$ 140,000[27,28]. Figure 1a is based on Singapore Department of Statistics' quinquennial Household Expenditure Survey, 2012/2013 HES for short[27]. The plot shows that the adoption of electricity and water-consuming appliances are consistent with the -S-curves reported in the literature[16,29,30]. In particular, the residential penetration of electricity-demanding AC is a relatively low 32% at the low end of the expenditure range, rising to 100% at the high end. At low levels of household expenditure, the adoption of washing machines already stands at a high 80%—let alone that of other water-using facilities such as showers, where adoption is near universal. At a given expenditure level, washing machines enjoy more widespread diffusion than AC (Fig. 1a). An observational study by the water agency of end uses in 2016/2017 found that showering accounts for 27% of Singapore's home water use, followed by bathroom tap/basin (18%), flushing (18%), kitchen (16%), and laundry (15%)[31].

Also based on the 2012/2013 HES, Fig. 1b shows that the electricity share of household expenditure falls less steeply as income rises compared with the water share of expenditure. This is consistent with the varying adoption of electricity versus water-consuming appliances at different points of the socioeconomic distribution; in particular, the wide variation in adoption of AC compared with showers and washers. Residential electricity and water prices, described below and reported over time in Fig. 1c, are essentially constant over quantity and hour of use, and are invariant across households. At local prices and adjusting income for household size[27], households in the bottom quintile of the income distribution spend 2.9 times more on electricity than on water (3.2% vs. 1.1% of expenditure), whereas households in the top income quintile spend 4.5 times more on electricity than on water (1.8% vs. 0.4%). In a tropical climate such as Singapore's, a key mechanism is likely to involve space cooling/ AC and showering/washing, as I subsequently discuss. (Water and electricity expenditure shares that decline in income are noted—though not compared—for US households in Timmins[32].)

To examine how residential water and electricity demand may respond differentially to heat across the socioeconomic distribution, I access a 40-month panel of water and electricity bills for a 1-in-10 random sample of households—namely, billed quantities for about 120,000 households from September 2012 to December 2015. Supplementary Table 1 describes the randomization procedure. The data provider, SP Services, provides one-stop metering and billing services to all households in Singapore[33]. Each dwelling has its separate metering facilities, and a dwelling's

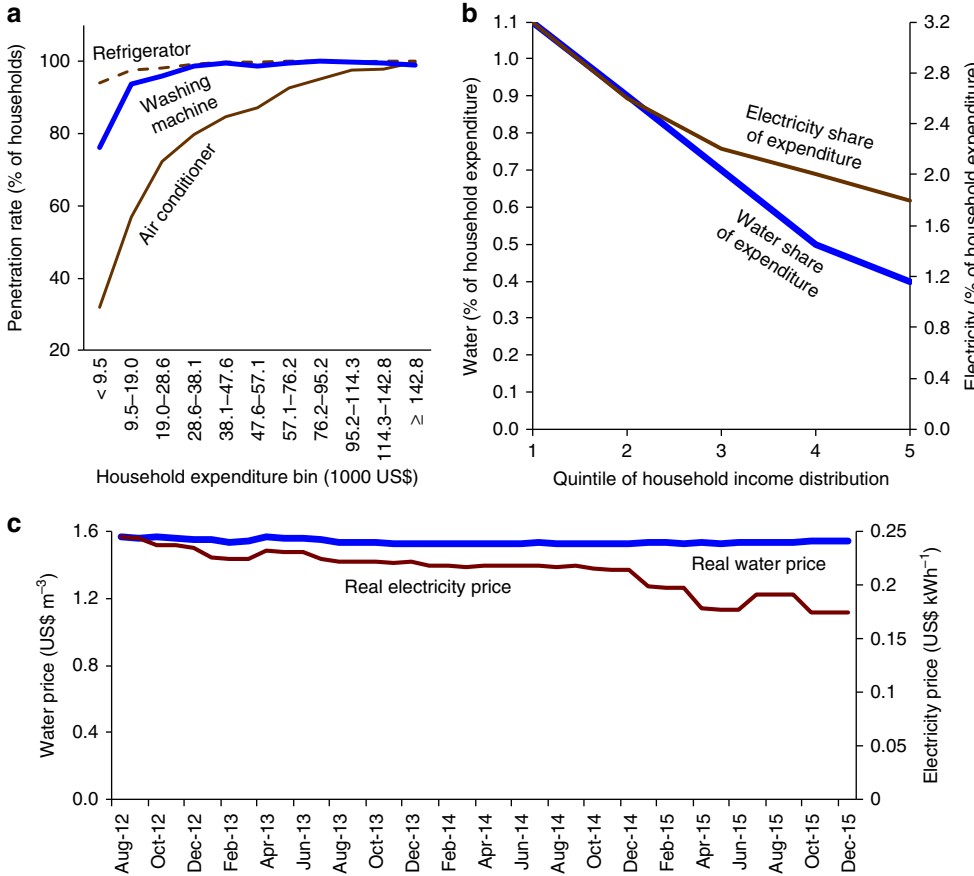

**Fig. 1** Appliance adoption and utility expenditure by income. **a** Household penetration rates (%) for air conditioners, washing machines, and refrigerators, by annual household expenditure bin (1000 US$ at 1.26 SG$ per 1 US$), in 2012/2013. The unit of study is a resident household in Singapore. **b** Water and electricity shares of household expenditure (%) by quintile of the distribution of household income, in 2012/2013. **c** Household water and electricity prices (in US$ m$^{-3}$ and US$ kWh$^{-1}$, respectively), 2012–2015, inclusive of a 7% General Services Tax and adjusted for changes in the Consumer Price Index (base June 2014). Water prices, for volumes up to 40 m$^3$ month$^{-1}$, include a conservation tax. Sources: 2012/13 Household Expenditure Survey, Energy Market Authority, Public Utilities Board. Source data are provided as a Source Data file

occupant holds an account with SP Services. A household, whether an owner-occupier or a renter, pays for the volume of water and electricity it uses. Singapore's home ownership rate is high across the income distribution, i.e., 84 and 90% in the bottom and top quintiles, respectively[27]. When a household moves to a new dwelling, it opens a new account with SP Services. Thus, for clarity, each account number corresponds to a unique combination of user and dwelling. By design (account number), the microdata control for—but do not track—a moving family over different dwellings. For simplicity, I use household to refer to a user by dwelling identifier in the microdata.

I restrict dwellings to apartment units, which account for 94% of Singapore's 1.3 million residential units[34], since water consumption among the relatively few families living in houses and bungalows is more likely to include confounding water demand for gardens[22]. The microdata contains a variable informing the dwelling type, to which I apply the apartment sample restriction. This variable directly informs on the household's apartment type, according to six standard labeled categories, ordered as follows: 1-room, i.e., a studio apartment; 2-room (typically 1 bedroom plus living room totaling about 50 m$^2$ floor area); 3-room (70 m$^2$); 4-room (90 m$^2$); 5 or 6-room (110 m$^2$) including executive (130 m$^2$); and condominium (a label for a unit in typically the most premium developments). Utilities consumed by any common areas in a condominium are billed separately to the condominium manager and do not contaminate

my sample. The microdata further informs the household's two-digit postal code (postal code for short) covering 73 geographic areas across Singapore.

The microdata do not inform on the individual household's income, size, or its adoption of appliances. It is typical that a utility or its metering and billing agent do not systematically collect socioeconomic characteristics of its customers[26]. In the absence of individual information on income, a household's apartment type serves as a proxy for its overall socioeconomic standing, as well as adoption of durable goods. The 2012/2013 HES[27] reports mean household income per capita and overall AC penetration by apartment type. Based on this source, Supplementary Table 2 report that both mean income per capita and AC adoption increase monotonically over the six apartment types ordered above, with ranges as wide as US$ 9300 person$^{-1}$ and 14% AC among 1-room apartment dwellers, to US$ 68,900 person$^{-1}$ and AC = 99% for condominium residents. Moreover, in a nation of owner-occupiers, residential property assets are a widespread component of wealth. Housing accounts for one-half of household net worth, and values typically increase over the six ordered apartment types[35–37]. Similarly, rent—whether paid out or imputed in the case of an owner-occupier—is a key component of household expenditure. Rent increases over the six ordered apartment types in line with their value (property prices reflect the present value of the flow of rent). Finally, a separate Household Interview Travel Survey that I subsequently bring into

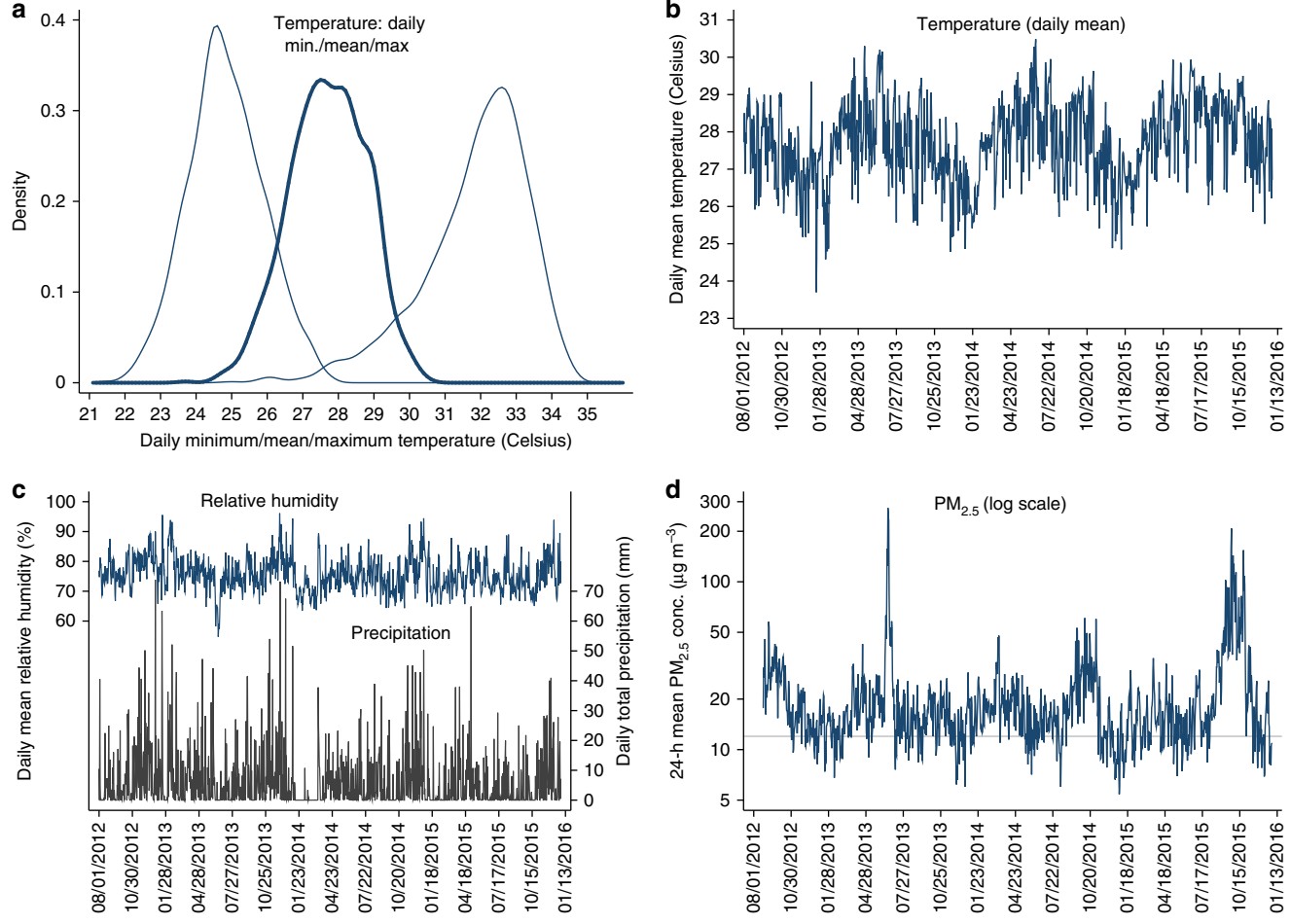

**Fig. 2** Ambient environmental conditions from 2012 to 2015. **a** Distribution of daily minimum, daily mean, and daily maximum temperature (°C). **b–d** Evolution of daily mean temperature (°C), daily mean relative humidity (%) and daily total precipitation (mm), and daily mean PM$_{2.5}$ mass concentrations ($\mu$g m$^{-3}$, in log scale), respectively. The sample period is August 1, 2012 (August 24, 2012 for PM$_{2.5}$) to December 31, 2015. An observation is a day. The horizontal line in **d** marks both the US EPA's primary standard and the Sustainable Singapore Blueprint target. Sources: Meteorological Service Singapore (temperature, precipitation), NUS Geography Weather Station (relative humidity), National Environment Agency (PM$_{2.5}$). Source data are provided as a Source Data file

the analysis reports a clear association between income and dwelling type (Supplementary Fig. 1).

Supplementary Table 3 describes weather and air quality variables (including sources) for Singapore over the study period, as potential environmental determinants of residential demand for utilities[9,14–16,38,39]. Daily mean temperature varies within a narrow range, with 5th and 95th percentiles of 25.9 and 29.4 °C (78.6 and 84.9 °F, Fig. 2a). Cooling degree days amount to 3500 °C y$^{-1}$ (at base temperature 18 °C). July is warmer than January, yet there is variation within time of year (Fig. 2b). Relative humidity hovers around 80% and precipitation, while variable, occurs throughout the year (Fig. 2c). Supplementary Fig. 2 reports stronger winds during the wetter Northeast Monsoon from December to early March than during the Southwest Monsoon from June to September. Residential demand for water and electricity may shift with air pollution through defensive behavior; for example, high (and visible) particle concentrations may induce households to stay at home or close the windows, thus flushing the toilet and using the AC more often. Daily mean PM$_{2.5}$ (particulate matter up to 2.5 $\mu$m in diameter) averages 21 $\mu$g m$^{-3}$. Twenty-four hour PM$_{2.5}$ routinely exceeds the US primary 1-year average National Ambient Air Quality Standard, and the nation's own Sustainable Singapore Blueprint target, of 12 $\mu$g m$^{-3}$ (Fig. 2d)[40,41]. PM$_{2.5}$ is nearly always

dominant in Singapore's Pollutant Standards Index, an air quality index based on the maximum value across subindices for the level of criteria pollutants SO$_2$, CO, O$_3$, NO$_2$, PM$_{10}$, and PM$_{2.5}$.

Among the strengths of the combined environment and residential indoor water/electricity panel and setting are its multiyear duration, enabling the control of seasonal determinants, and its individual household level, a feature that, compared to cross-sectional designs, allows one to correct for unobserved heterogeneity. This includes differences in user behavior (e.g., a preference for long showers, family composition) and property characteristics (e.g., west-facing apartments with exposure to the afternoon sun, dual-flush toilets). As for time-varying determinants other than environment and season, Singapore's stable economy over the study period suggests that any unobserved shock to household demand that might correlate with de-seasoned temperature variation is likely to be small. For example, the 2012–2015 average resident unemployment rate hardly varied, between 2.7 and 2.8%[42]. In newly affluent Singapore, and despite incomes that vary widely, residential water and electricity connections are near universal. Supply disruptions are rare[43], with water reliably sourced from local catchments, Malaysia, and desalination; electricity is generated from natural gas. There has been no discernible variation in household

resource conservation measures or public information campaigns, even during the early 2014 dry spell in the case of water[44].

The relative simplicity and invariance of prices in my setting is also helpful. A large economics literature estimates the price elasticity of household water and electricity demand[19], but here prices are not the object of interest; rather, they are a variable to be controlled for. In fact, prices have either not changed (water) or trended downward (electricity) (Fig. 1c). For water and electricity alike, all households face the same marginal price schedule. Marginal prices are either constant (electricity) or essentially constant (water) over quantity supplied. Water is charged at an increasing block price, as in Hewitt and Hanemann[21,45], but only 4% of my sample reaches a second block of $40 \, m^3 \, mo^{-1}$, where marginal price is 20% higher. This second block may matter more to the relatively few residential properties that are excluded from my sample—namely, houses and bungalows, due to landscape irrigation.

Supplementary Table 1 describes a water-to-electricity quantity ratio in the microdata—in arbitrary $m^3 \, kWh^{-1}$ and averaged across household by period observations within apartment type—that decreases monotonically over the six ordered apartment types, from 0.113 for 1-room apartments compared to a lower 0.033 for condominium apartments. This pattern is in agreement with the water and electricity burdens reported in the 2012/2013 HES (Fig. 1b). The microdata and 2012/13 HES, summarized in Supplementary Tables 1 and 2, respectively, jointly indicate that average electricity use in lower-income, mostly AC-free 1-room apartments is 21% that of high-income, AC-saturated condominium apartments (134 against $622 \, kWh \, mo^{-1}$). In comparison, water use in 1-room apartments is 59% that of condominium apartments (9.8 against $16.5 \, m^3 \, mo^{-1}$). The 2012/2013 HES further indicates that average occupancy in 1- or 2-room apartments is 62% that of condominium apartments (2.1 against 3.4 persons).

Apartments with up to three rooms make up 25% of the random sample's observations, yet account for 20% of aggregate water consumption vs. 15% of aggregate electricity consumption. At the high end of the income distribution, the condominium apartments that comprise 17% of observations account for 16% of aggregate water consumption vs. 25% of aggregate electricity consumption. This pattern is informative in a difference-in-difference sense: smaller apartments' water versus electricity shares of aggregate use relative to larger/more premium apartments' water versus electricity shares. The microdata indicate that electricity use increases more steeply over the socioeconomic distribution than does water use. In fact, taking each apartment type and dividing mean resource use in the microdata by mean household size by apartment type obtained from the 2012/2013 HES[27], water use per person *decreases* over the socioeconomic distribution, while electricity use per person increases (Supplementary Table 2).

How comparable are Singapore's residential water outcomes to those elsewhere? Median consumption and price in a 2008 survey of households across OECD countries, which included houses, was $12 \, m^3 \, mo^{-1}$ and $1.33 \, € \, m^{-3}$ [23]. Against these numbers, Singapore apartments' median water use is 25% higher ($15 \, m^3 \, mo^{-1}$), and the residential water price is similar (1.54 US$ $m^{-3}$). A median household size of three persons in the OECD survey is comparable to that in Singapore. Average water demand by Korean households in 1982, of $17 \, m^3 \, mo^{-1}$ in $43 \, m^2$ flats and $26 \, m^3 \, mo^{-1}$ in $105 \, m^2$ flats, is higher than present-day use in Singapore's similarly sized 2-room and 5-room apartments: respectively, 13 and $19 \, m^3 \, mo^{-1}$ [46]. More recent consumption in 2000 for flats and apartment units (mean occupancy 1.4 persons) in Adelaide City, a medium-density area of Adelaide, Australia, averaged $6 \, m^3 \, mo^{-1}$, compared with $11 \, m^3 \, mo^{-1}$ in Singapore's

1–2 room apartments (occupancy 2.1 persons)[47]. A 6-year panel study in Santa Barbara County, CA, reported an insignificant (negative) association between household income and water use in high-density houses, for which landscape irrigation demand is low[13]. The study collected many household and property variables on a relatively small sample of 119 households, but did not survey electrical appliances or electricity use.

**Residential water and electricity demand respond to heat.** Table 1 reports the coefficient on average temperature in regressions of the natural logarithm (log for short) of water use in panel A, and of log electricity use in panel B, as I progressively add exogenous and endogenous controls (see Methods). Standard errors, reported in parentheses under point estimates, are clustered by household. This allows unobserved demand determinants to correlate over time within household, with 19 periods being the modal number of observations per household (Supplementary Fig. 3). The table reports point estimates and standard errors in log points, for a +1 °C variation. Taking the log of resource use as the dependent variable yields temperature coefficients that have the approximate interpretation of a percent increase over the household's baseline use, whether this use is high for a big household or low for a small household. Figure 3 presents the same results converted to 95% confidence intervals of percentage increase in resource use (+0.01 log points in use is equivalent to a $e^{0.01} - 1 \cong 1.0\%$ increase, further noting that 95% CI = point estimate ± 1.96 times the standard error).

In specification 1, with no controls other than a full set of household fixed effects—about 120,000 intercepts, also included in all other specifications—a 1 °C (1.8 °F) rise in average temperature during the usage period is associated with statistically significant increases of 1.1% in water use and 10.8% in electricity use. The empirical support for average-period temperature is 26.1–29.1 °C across 2 million household by period observations. Estimates vary somewhat as I add time and weather controls in specifications 2 and 3, respectively. With the inclusion of average $PM_{2.5}$, in specification 4, a 1 °C temperature rise increases water use by 0.7% and electricity use by 7.8%. Specification 5 replaces year fixed effects with a linear time trend, and specification 6 allows $PM_{2.5}$ to be endogenous, with a two-stage least squares (2SLS) estimator yielding estimates that are similar to ordinary least squares (OLS). The lower temperature effect in specifications 4 and 6 relative to specification 3 illustrate the importance of controlling for $PM_{2.5}$ when examining the weather determinants of residential resource use. This is an important point that merits highlighting. Moreover, I find that a $10 \, \mu g \, m^{-3}$ increase in average $PM_{2.5}$ during a usage period increases water demand by 0.5% (omitted from Table 1 for brevity). Controlling for the incidence of rain, conditions that are more humid—higher relative humidity, lower dew point depression—raise water demand.

Figure 4/Supplementary Table 4 repeats all six specifications of Fig. 3/Table 1, except that the variable of interest enters the resource demand equation not as an average but as the proportion of days over the usage period in which the daily mean temperature exceeded 28.5 °C. The empirical support for the proportion of usage period with above 28.5 °C, days variable is 0–83%. Days with 24-h means above 28.5 °C account for one-quarter of the days in the sample and exhibit a maximum temperature, on average, of 33.0 °C. In specification 4, raising the proportion of above 28.5 °C days from 0 to 100% of a usage period would raise water use by 2.2%. Supplementary Table 4 provides sensitivity analysis by taking the proportion of above 29 °C days as a measure of hotter usage periods. Only 12% of days in the study period exceed this threshold. Given Singapore's

**Table 1 Household demands for water and electricity respond to heat**

| Point estimate (and standard error in parentheses) are expressed in log points | (1) Only HH FE, OLS | (2) with time, OLS | (3) with weather, OLS | (4) with PM$_{2.5}$, OLS | (5) Trend, not year, OLS | (6) PM$_{2.5}$ endogenous, 2SLS |
|---|---|---|---|---|---|---|
| *Dependent variable: Log water use (m$^3$ mo$^{-1}$)* | | | | | | |
| Mean temperature over usage period (+1 °C) | 0.0106*** | 0.0124*** | 0.0106*** | 0.0067*** | 0.0049*** | 0.0072*** |
| | (0.0004) | (0.0014) | (0.0015) | (0.0016) | (0.0016) | (0.0015) |
| Number of observations | 2,001,389 | 2,001,389 | 1,998,127 | 1,998,127 | 1,998,127 | 1,998,127 |
| Number of regressors (not counting HH FE) | 1 | 23 | 27 | 28 | 26 | 28 |
| Number of households | 121,480 | 121,480 | 121,480 | 121,480 | 121,480 | 121,480 |
| $R^2$ [first-stage $F$-statistic, excluded instr.] | 0.830 | 0.830 | 0.830 | 0.830 | 0.830 | [926,239] |
| Mean value of usage in sample (m$^3$ mo$^{-1}$) | 16.66 | 16.66 | 16.66 | 16.66 | 16.66 | 16.66 |
| *Dependent variable: Log electricity use (kWh mo$^{-1}$)* | | | | | | |
| Mean temperature over usage period (+1 °C) | 0.1026*** | 0.0987*** | 0.0828*** | 0.0751*** | 0.0738*** | 0.0746*** |
| | (0.0004) | (0.0009) | (0.0010) | (0.0010) | (0.0010) | (0.0010) |
| Number of observations | 1,982,472 | 1,982,472 | 1,979,238 | 1,979,238 | 1,979,238 | 1,979,238 |
| Number of regressors (not counting HH FE) | 1 | 23 | 27 | 28 | 26 | 28 |
| Number of households | 120,099 | 120,099 | 120,099 | 120,099 | 120,099 | 120,099 |
| $R^2$ [first-stage $F$-statistic, excluded instr.] | 0.885 | 0.885 | 0.885 | 0.886 | 0.886 | [930,901] |
| Mean value of usage in sample | 405.91 | 405.91 | 405.86 | 405.86 | 405.86 | 405.86 |

The table reports estimates for 12 water or electricity use regressions. An observation is a household by usage period in the 2012–2015 utility usage microdata. The dependent variable is the natural logarithm (log) of water use (m$^3$ mo$^{-1}$) in panel **a** and log electricity use (kWh mo$^{-1}$) in panel **b**. The key regressor is the average daily mean temperature (°C). This average is taken over the same days that are concurrent to each usage observation, and has empirical support 26.1–29.1 °C. All regressions include a full set of household fixed effects (FE). Relative to specification 1, specification 2 includes time controls (month FE, year FE, day type composition; see Methods). Relative to 2, specification 3 includes weather controls (average daily mean relative humidity, dew point depression, wind speed; proportion of days with some precipitation). Relative to 3, specification 4 controls for average daily mean PM$_{2.5}$. Relative to 4, specification 5 replaces year FE with a linear time trend. Relative to 4, specification 6 instruments for PM$_{2.5}$ with Southeast Asia fire activity variables and Singapore thermal gradients and wind direction. OLS regressions in columns 1–5, 2SLS regressions in column 6. Standard errors, in parentheses, are clustered by household. ***, **, * denote significance at the 0.01, 0.05, and 0.1 levels, respectively.

tropical weather, the daily mean temperature did not exceed 30.5 °C in the study period.

As a further sensitivity analysis, Supplementary Fig. 4 presents estimates for OLS regression models of water and electricity use, with average temperature entering nonlinearly via highly granular bins of width 0.4 °C. The linearity restriction imposed in Fig. 3 is broadly borne out by the data.

**Heat impacts in the socioeconomic cross-section.** Taking specification 4 of Fig. 3 as the point of departure, I consider interactions between the variable of interest and indicators for the different apartment types, as proxies for different socioeconomic groups. This allows estimated heat impacts to differ by group. Six groups is not low in applied economics research; e.g., Goolsbee and Petrin[48] choose to model household demand using five discrete income groups (and despite observing a continuous measure of income). I also allow interactions between apartment-type indicators and specific time controls, namely year fixed effects and school holidays (Methods). Figure 5a plots apartment-specific water against electricity demand responses to +1 °C.

I find that the percentage water demand response to heat at this tropical location is stronger among households in 1-room apartments—i.e., a statistically significant 2.6% increase (per +1 °C)—compared to a statistically insignificant 0.1% increase among condominium apartments. A test of equal water responses is rejected with a p-value of <0.0001 (Supplementary Table 5 reports all equality tests among pairs of apartment types). In sharp contrast, the electricity demand increase per +1 °C is largest among households in AC-equipped higher-priced condominium apartments, i.e., +9.6%, compared to a statistically significantly different +4.2% in mostly naturally ventilated 1-room apartments. Emphasizing the wide range of socioeconomic variation, Fig. 5b plots AC penetration and mean household income per person across apartment types (obtained from the 2012/2013 HES). AC adoption and household income per person are each seven times higher among condominium apartments than among 1-room apartments.

Figure 5 shows that the percentage water response to heat decreases monotonically from low to high socioeconomic groups, in marked contrast to the electricity response, which increases monotonically. Exposed to heat, 1-room and 2-room apartments increase water use more relative to their baseline consumption than five-room and condominium apartments, where the electricity response is disproportionately larger. The evidence shows that adaptation to routine temperature variation differs across the city–state's heterogeneous income groups, and this correlates with adoption of residential AC.

Given the narrow range of temperature variation in the tropics compared to a temperate climate such as Europe or North America over the seasons, Fig. 5a's restriction that resource demand-temperature slopes be constant over temperature is appropriate. Importantly, I allow these slopes to vary with apartment type. A similar result obtains if I restrict resource demands in levels—rather than their natural logarithm, as in the analysis thus far—to be linear in temperature (Supplementary Fig. 5b). Supplementary Fig. 6 illustrates the relative importance of different environmental controls. Taking the Fig. 5a specification as the starting point for each case, I re-estimate the resource demand models dropping first, relative humidity and dew point depression, second, wind speed, and third, PM$_{2.5}$. I find that in each case, heat effects change somewhat, for example, rising in magnitude when not controlling for PM$_{2.5}$, and shrinking in magnitude when not controlling for humidity. I describe further sensitivity analysis below.

**Heat-water response declines with average electricity use.** The microdata show substantial variation between the 73 geographic areas (postal codes) in their average electricity demand per household (within apartment type). For 1-room apartments, for example, the distribution of average electricity use across postal codes exhibits 5th and 95th percentiles of 109 and 150 m$^3$ mo$^{-1}$ household$^{-1}$. For condominium apartments, 5th and 95th percentiles across postal codes are 468 and 894 m$^3$ mo$^{-1}$ household$^{-1}$.

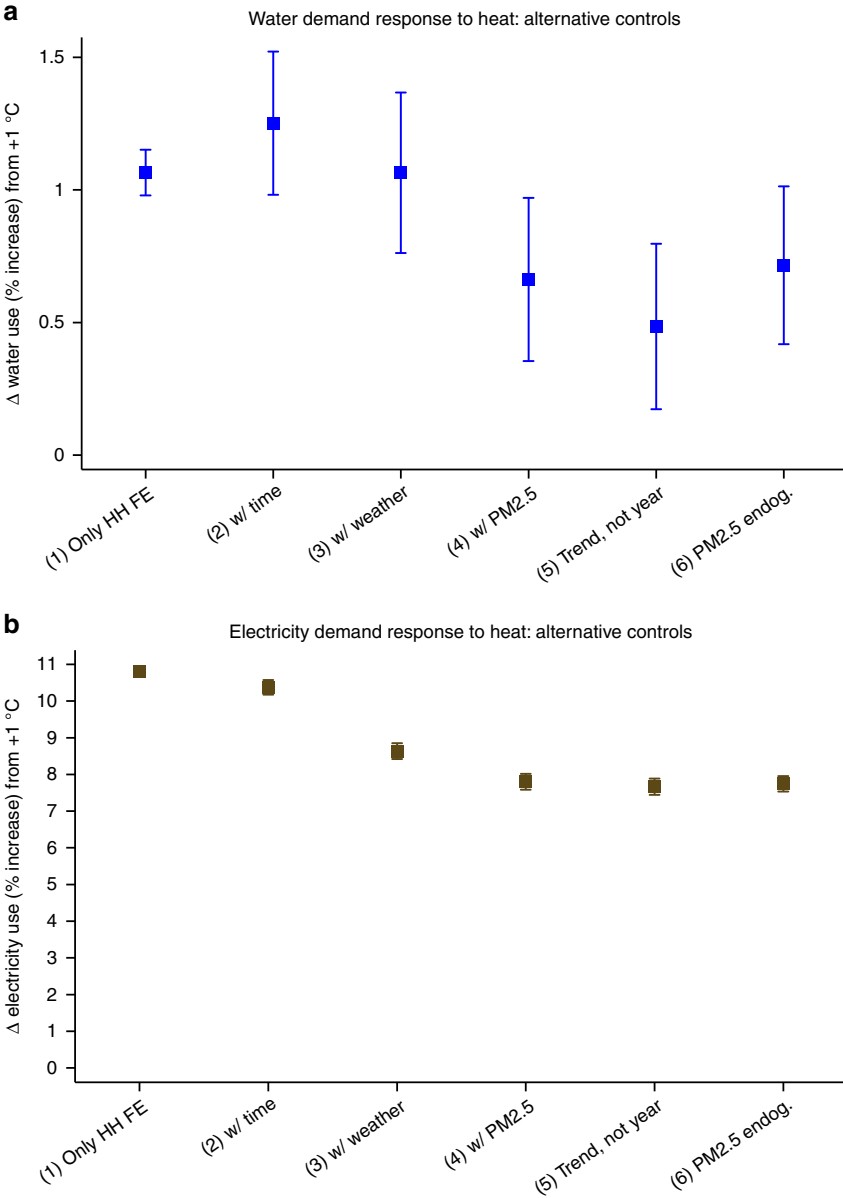

**Fig. 3** Household demands for water and electricity respond to heat. 95% Confidence intervals for the percent increase in **a** water use and **b** electricity use, from a +1 °C variation in the average daily mean temperature—the key regressor. All specifications include a full set of household fixed effects. Relative to specification 1, specification 2 includes time controls. Relative to 2, specification 3 includes weather controls. Relative to 3, specification 4 controls for average daily mean $PM_{2.5}$. Relative to 4, specification 5 replaces year fixed effects with a linear time trend. Relative to 4, specification 6 instruments for $PM_{2.5}$ with Southeast Asia fire activity variables and Singapore thermal gradients and wind direction

This is likely to reflect, at least in part, geographic differences in electrical appliance holdings, such as AC, even for the same apartment type. To the water demand specification 4 of Fig. 3/ Table 1, I include an interaction between temperature and (time-invariant) average electricity use per household in the household's apartment type within geographic area. Supplementary Table 6 documents that the magnitude of the positive heat-water response is lower among households in local markets—geographic area by apartment type—in which per household average electricity use is higher.

**Heat-water response declines with average income**. A separate 2008 Household Interview Travel Survey (HITS), conducted by the Land Transport Authority and used in Fesselmeyer and Liu[49], contains income as stated by individual members of 9532 households residing in apartments across 63 (two-digit) postal

codes across Singapore. Also reported are the respondent's apartment type and postal code. (This excludes 203 households that refused to declare income.) Monthly personal income is reported in 12 mutually exclusive bins.

From the HITS, I take each combination of geographic area by apartment type—again, a market—and compute the proportion of households in which at least one member states earning over US\$ 1984 $mo^{-1}$, e.g., a household with two parents and two children in which one parent selects the (local currency) SG\$ 2500–2999 bin or higher. Across the 63 postal codes covered in the HITS, the median proportion of households with over US\$-1984 individuals is zero for 1-room apartments, compared to 0.67 for condominium apartments. For each apartment type, Supplementary Fig. 1 reports other quantiles across postal codes for the proportion of households with such high earners; income and apartment type show a clear correlation.

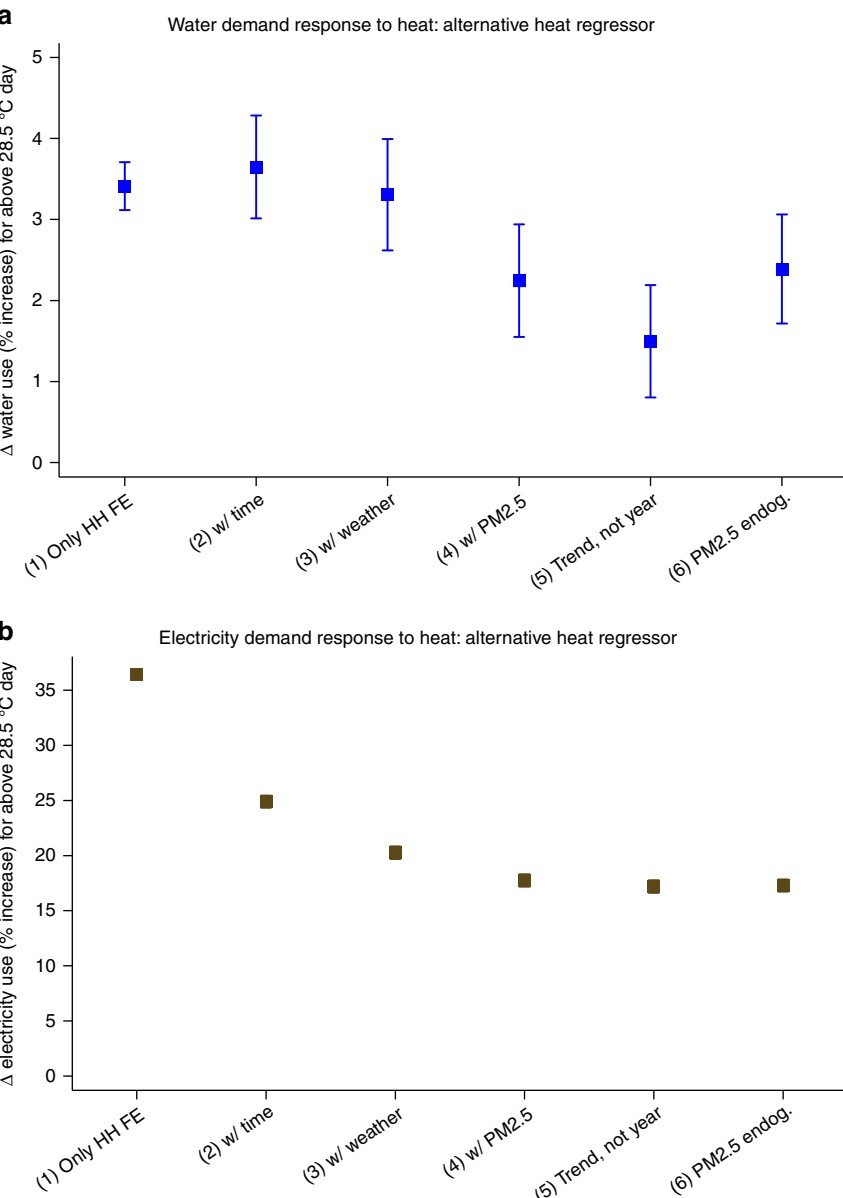

**Fig. 4** Alternative heat covariate in the household demand model. 95% Confidence intervals for the percent increase in **a** water use and **b** electricity use, from a 0 to 100% variation in the proportion of days with daily mean temperature above 28.5 °C—an alternative key regressor. All specifications include a full set of household fixed effects. Relative to specification 1, specification 2 includes time controls. Relative to 2, specification 3 includes weather controls. Relative to 3, specification 4 controls for average daily mean $PM_{2.5}$. Relative to 4, specification 5 replaces year fixed effects with a linear time trend. Relative to 4, specification 6 instruments for $PM_{2.5}$ with Southeast Asia fire activity variables and Singapore thermal gradients and wind direction

To the water demand specification 4 of Fig. 3/Table 1, I include an interaction between temperature and this HITS market-level income measure (or an interaction between temperature and an over US$-1190 income measure, as an alternative). Table 2 shows that the magnitude of the positive heat-water response is lower in markets (apartment type by geographic area) with more affluent households. Since the number of HITS respondents by market ranges from 1 to 249 households, I alternatively weight regressions by the square root of the number of respondents to control for sampling error in the income measure.

**Robustness**. Across apartment types, a joint pattern is robust across several alternative specifications. First, the heat-water response is inversely associated with income and AC adoption. Second, the heat-electricity response is positively associated with income and AC adoption. Relative to Fig. 5a's specification with

apartment-type interactions, I alternatively drop average dew point depression while keeping relative humidity in the vector of weather controls (Supplementary Fig. 7a); enter all weather and $PM_{2.5}$ controls via bins rather than linearly (Supplementary Fig. 7b); or include interactions of the real electricity price and apartment-type indicators as controls in the electricity equation (Supplementary Fig. 7e). In the latter robustness test, I take electricity prices to be exogenous, following the Energy Market Authority's policy of updating tariffs quarterly to reflect changes in the cost of power generation[50]. Results are similar if instead I assume electricity prices are endogenous to demand shocks and take a proxy for world natural gas prices—the key cost-shifter—as an excluded instrument[51], estimating the model by 2SLS (estimates omitted for brevity).

To account for time-varying household income beyond year fixed effects, I control for the inflation-adjusted overall unit labor

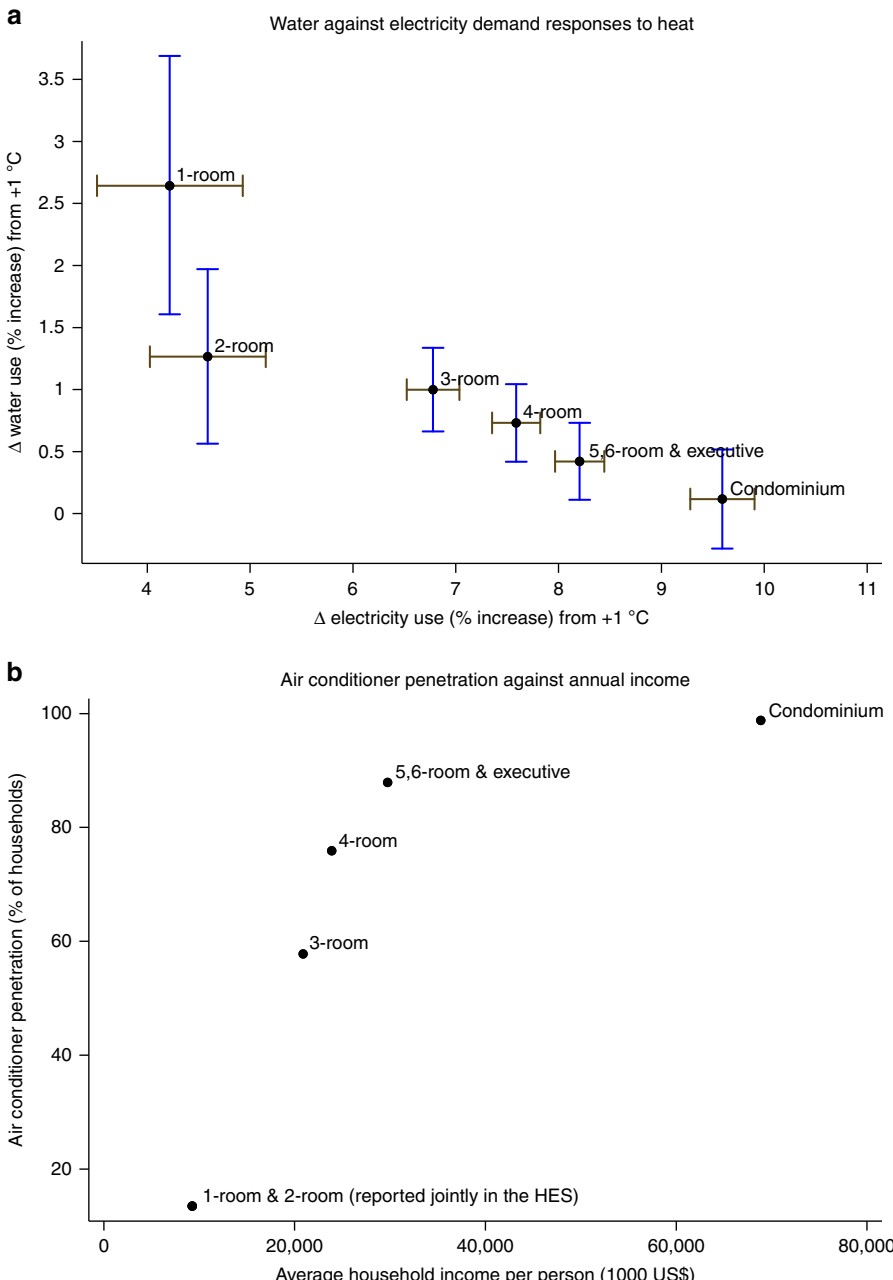

**Fig. 5** Water versus electricity demand responses to heat by type. **a** 95% Confidence intervals for the percent increase in water demand against the percent increase in electricity demand, by apartment type, from a +1 °C variation. The variable of interest, average temperature over the usage period, enters flexibly interacted with a set of indicators for the different apartment types. The empirical support for average temperature is 26.1–29.1 °C. The regression models control for household, time (month, year, day-of-the-week, public holidays, and school holidays), weather (relative humidity, dew point depression, wind speed, and precipitation), and $PM_{2.5}$. I specify interactions between apartment-type indicators and controls for year and school holidays. **b** Air-conditioner penetration (%) and mean annual household income per person (1000 US$) by apartment type, reported in Table 53 and Chart 2.2c of the 2012/2013 Household Expenditure Survey. Source data for panel **b** are also provided as a Source Data file

cost or the average wage in the economy. In either case, I include interactions between the labor market index and apartment-type indicators to allow for varying income sensitivity (Supplementary Fig. 7c, d)[52,53]. Outdoor $PM_{2.5}$ may affect the time households stay at home, or the electrical appliances they use, and this averting behavior can vary by income group[54,55]. Therefore, to account for such $PM_{2.5}$-induced shifts in resource use, I include interactions between $PM_{2.5}$ and apartment-type indicators (Supplementary Fig. 7f). As an alternative to average temperature

in the usage period, I specify the proportion of days with daily mean temperature above 28.5 °C, including interactions between this measure of heat stress and apartment-type indicators (Supplementary Fig. 5a). I implement models of residential resource use very flexibly by apartment subsample, with loss of precision (Supplementary Fig. 5c). I estimate Fig. 5a's specification by 2SLS instead of OLS, allowing $PM_{2.5}$ to be endogenous to unobserved demand shocks (Supplementary Fig. 8a, b). A further robustness test allows for possible measurement error in the main

**Table 2 Heat-water response is lower in markets with higher average income**

| Market-level income by temperature interaction. Point estimates and std.err are expressed in log points | (1) $1984, unweighted | (2) $1984, weighted | (3) $1190, unweighted | (4) $1190, weighted |
|---|---|---|---|---|
| *Dependent variable: log water use* | | | | |
| Mean temperature over usage period (°C) | 0.0091*** | 0.0100*** | 0.0120*** | 0.0118*** |
| | (0.0020) | (0.0018) | (0.0026) | (0.0025) |
| Mean temperature (°C) × market-level proportion of HHs with over-US$-1984 individuals (%) | −0.0084*** | −0.0089*** | | |
| | (0.0025) | (0.0024) | | |
| Mean temperature (°C) × market-level proportion of HHs with over-US$-1190 individuals (%) | | | −0.0098*** | −0.0083*** |
| | | | (0.0028) | (0.0027) |
| Weigh regression by square root of respondent HHs? | No | Yes | No | Yes |
| Number of observations (114,866 HHs) | 1,900,356 | 1,900,356 | 1,900,356 | 1,900,356 |
| Mean value of usage in sample (m³ mo⁻¹) | 16.86 | 16.86 | 16.86 | 16.86 |

The table reports estimates for four water use regressions. An observation is a household by usage period in the 2012–2015 utility usage microdata. The dependent variable is log water use. The regressors of interest are average daily mean temperature (°C) and its interaction with the income measure for the household's (HH) postal code and apartment type (local market). This market-level income measure is the proportion of households where at least one member states earning over a specified cutoff, computed by postal code by apartment type from a separate 2008 Household Interview Travel Survey (HITS). The monthly personal income cutoff is US$ 1984 in specifications 1 and 2, and US$ 1190 in specifications 3 and 4; both income measures have empirical support 0–1 and are missing for some markets. All regressions include controls for household, time (month, year, day-of-the-week, public holidays, and school holidays), weather (relative humidity, dew point depression, wind speed, and precipitation), and PM2.5. Average market-level income is included but subsumed in the household fixed effects. OLS regressions. Standard errors (std.err), in parentheses, are clustered by household. ***, **, * denote significance at the 0.01, 0.05, and 0.1 levels, respectively.

variable of interest, temperature (Supplementary Fig. 8c, test described in Methods). Further testing the linear-in-temperature restriction in Fig. 5a, I include interactions between the square of temperature and apartment-type indicators (Supplementary Fig. 9a), and, additionally, the square of each environmental control (Supplementary Fig. 9b).

## Discussion

Evaluated at a mean water use of 9.8 m³ mo⁻¹ for 1-room apartments, a 2.6% (95% CI: 1.6–3.7%) increase in water consumption due to a +1 °C heat shock translates into an additional 8.6 L day⁻¹ for the household. Survey evidence for Singapore quantifies mean water use per shower at 20 L[56], so the magnitude of the response to +1 °C is equivalent to one additional shower for every 2.3 households. This magnitude is also significant when judged by the response to experimental interventions in the resource conservation literature[57]. Given a mean household size of 2.1 persons, the heat-relief behaviors that underlie this demand shift likely include increased washing of body and clothing, and drinking liquids. Other cooling behaviors are conceivable. For example, given widespread residential diffusion of refrigerators and relative humidity averaging 76%, one mechanism for achieving cooling without the need for evaporation could be to use the refrigerator to cool wet towels that one then places on the body. A 300-person survey targeted at the general population suggested that the water-based cooling behaviors that are prevalent in Singapore residences take place in the shower (used more often or for longer), wash basin, and laundry (Methods and Supplementary Table 7).

The microdata are limited in the sense that they do not detail consumption by end-use category. Future work can use metered devices to investigate the precise mechanisms by which household water demand, for a lower-income group of about US$ 10,000 person⁻¹—and mostly pre-AC adoption—responds to heat shocks currently observed in the tropics. For example, one can attach time-resolved meters that record both the intensive margin (resource flow) and the extensive margin (on/off) directly to the shower and AC units. By observing a household's income, $m$, and AC adoption and use, modeled as $AC(m, p_{AC}, T)$[16,58], one can then estimate water demand, $q_W(m, p_W, T, AC)$, as part of a system, with $p_W$ and $p_{AC}$ further denoting the prices of water and AC

(adoption and use). Shifts in temperature, $T$, impact water demand both directly and indirectly via AC.

As incomes and AC adoption increase across other apartment types, the +1 °C-induced shift in water demand, while positive, declines in magnitude, namely: +5.3 L day⁻¹ for 2-room apartments; +4.8 L day⁻¹ for 3-room; +4.3 L day⁻¹ for 4-room; +2.7 L day⁻¹ for 5-room; and +0.6 L day⁻¹ and statistically insignificant for condominium apartments. Since occupancy tends to grow from 1-room to 5-room apartments, the drop in the magnitude of the per capita water demand increase is even steeper. One interpretation of this result is that energy-intensive AC can substitute for water-based cooling services, and there are systematic, predictable differences in income and AC adoption (on average) across apartment types (Fig. 5b and S10). An alternative interpretation would be that of a convergence response, with high-income individuals showering at will—say, two daily showers irrespective of heat—whereas cost-sensitive low-income individuals would raise their water use—to a bliss point of two daily showers, say—only during hot weather. Both interpretations are consistent with the evidence presented in Fig. 5a and Table 2 that population subgroups with lower income, lower (housing) wealth, and/or lower AC access increase their water consumption in response to routine heat. While testing between these and other alternative interpretations is beyond the scope of this research, the pattern documented in Supplementary Table 2— that indoor water use, after adjusting for differences in mean household size, on average declines from 1-room to condominium apartments—does not seem to favor the convergence hypothesis.

In sharp contrast to the ordering of the water demand response, the electricity demand response to heat stress is maximal among condominium apartments, with a 9.6% (95% CI: 9.3–9.9%) increase per +1 °C translating into an additional 2.0 kWh day⁻¹. According to the National Environment Agency[59], median power consumption across 400 AC models sold in Singapore is 9400 kWh y⁻¹ when operated continuously and subject to typical weather and thermostat settings. By comparison, the median across 500 refrigerator models is 420 kWh y⁻¹. A +2.0 kWh day⁻¹ electricity demand response to +1 °C then amounts to running an AC unit, which 99% of condominium apartments have access to, for an additional 2 h per day. This subgroup's high income and AC saturation may explain why the temperature response for this

climate zone is about double that typically estimated for a US population[14,15,18,60,61]. To add further context, a study of Singapore school children equipped with portable sensors estimated that median (resp., mean) AC exposure over a 24-h cycle is 4 h (resp., 6 h)[62].

Across the other apartment types, the +1 °C-induced shift in electricity demand rises concomitant with income and AC adoption, namely: +0.2 kWh day$^{-1}$ for 1-room apartments; +0.3 kWh day$^{-1}$ for 2-room; +0.6 kWh day$^{-1}$ for 3-room; +1.0 kWh day$^{-1}$ for 4-room; and +1.3 kWh day$^{-1}$ for 5-room. These electricity demand shifts grow more steeply over apartment types than baseline use (the latter is partly driven by differences in occupancy), resulting in *percent* increases (Fig. 5a). Differences in the adoption and use of refrigerators, unlike AC, do not seem significant enough to explain the heat-induced shift in electricity demand over the socioeconomic distribution. The penetration of refrigerators already stands at a high 93% among Singapore's lower-income households in 1-room and 2-room apartments. While high-income households may own larger and newer refrigerators than low-income households, and cooling requires more power in hot weather, larger and newer refrigerators tend to be more energy efficient per unit of capacity.

Valuing consumption changes at average residential utility prices, a +1 °C shock increases an average 1-room apartment's monthly expenditure by US\$ 0.40 and US\$ 0.98 on water and electricity, respectively. In warmer weather, households may demand more electricity to power fans—in addition to AC for the 14% of 1-room apartments with AC—as well as refrigerators, which require more power to maintain a reference temperature. At the other end of the socioeconomic distribution, a +1 °C shock raises a condominium apartment dweller's monthly water and electricity expenditure by US\$ 0.03 (not statistically significant) and US\$ 10.41, respectively.

The US Department of Energy has called for increased integration in the analysis and development of interconnected water and energy systems—the so-called water energy nexus—most notably from an infrastructure perspective, including the energy consumed in supplying water[63]. The agency argues that climate change, including shifting temperature extremes and rainfall variability, strengthens the case for a coupled approach to regulating the use of strained water and energy resources, including associated greenhouse gas emissions. One can view this research as taking the water-energy nexus to the household domain, particularly in the context of an urbanizing tropical landscape undergoing both climatic and human transitions. Documenting a water-energy nexus in household heat-relief amenities, which varies by socioeconomic group, is this article's key contribution. It adds to a nascent literature that examines temperature-dependent consumption patterns focused largely on North America and Europe.

Further work should investigate whether the empirical findings for Singapore extend to other urban populations in differing climates and/or levels of development, and thereby help improve real-time demand forecasting for multi-utilities. For example, other cities in tropical Asia experience similar climates, but lag Singapore in terms of economic development[64]. As in Singapore, incomes in these cities are distributed over a wide range, but with greater density at the lower end. Mean household size is higher in Manila and Kuala Lumpur (4 persons) but similar in Bangkok and Ho Chi Minh City (3). Residential AC adoption ranges from about 30% (Jakarta, Manila) to 50% (Kuala Lumpur, Ho Chi Minh City). Residential grid connections can be nearly universal (Bangkok). To a varying degree, urban water availability shapes urban development, as has been the case for Singapore, at least over the past half century[44]. From 1948 to 2016, annual mean temperature rose at a rate of 0.25 °C per decade, and the city-state

has invested heavily in new water sources, including desalination and reuse[65].

Observational evidence on multi-utility demand responses to heat by Singaporeans can shed light on how other populations, in tropical Asia and beyond, will respond as incomes rise and the climate warms. In Southeast Asia, climate models project annual temperature to increase by 1–4 °C and winter rainfall to decrease by 20–30% by 2070[64]. Only 8% of the 3 billion people living in the tropics currently have AC, compared to over 90% in the US and Japan[25]. One local policy implication for water-stressed cities that are host to a sizable mass of lower-income, pre-AC adoption households may be to tilt prices in favor of AC use, inducing a drop in water use. Reduced water demand adds to the health, productivity, and other socioeconomic benefits of AC use documented in the literature[60,66–70]. For example, AC has been credited with a decline in heat mortality in the US over the past century as well as lower heat mortality in the US than in India (for comparable climate zones)[67,68]. Even in Europe, the press has attributed fewer excess deaths during 2018's hot summer to increased AC access, compared to the 2003 heat wave; research should investigate these variables along with water demand[25]. At the same time, policies that promote AC need to weigh impacts on the aerosol burden and greenhouse emissions from electricity generation and common refrigerants such as hydrofluorocarbons.

## Methods

**Empirical models of household resource demands.** I estimate models of residential water (analogously, electricity) demand that take the form:

$$\text{water}_{it} = f(T_t, \lambda_i) + X_{it}\alpha_i + \phi_i + \delta_t + \varepsilon_{it} \tag{1}$$

An observation is an individual household $i$ by period $t$ of usage observed between two actual meter readings. This estimating equation relates the natural logarithm of utility use, $\text{water}_{it}$ (or $\text{electricity}_{it}$), to observed drivers, $(T_t, X_{it}, \phi_i, \delta_t)$, as well as unobserved determinants, $\varepsilon_{it}$, of demand. Observed demand determinants, further described below, include ambient temperature $T_t$, which enters the relationship according to a potentially nonlinear and household-specific function $f_i(.)$; other potential environmental drivers $X_{it}$; time-invariant shifters that differ by household $\phi_i$; and time-varying variables $\delta_t$. $(\lambda_i, \alpha_i)$ are parameters that govern the relationship between environmental conditions and resource demand.

With regard to household by period units of observation, the vast majority of periods have a bimonthly duration. SP Services bills customers every month, but only about one-half of these bills are based on actual readings, which necessitate (per the current processes and technology deployed) that an SP Services staff physically visit the dwelling to read water and electricity meters. These meter-reading visits to dwellings are undertaken once every 2 months, and the interim monthly bill is based on estimated use[71]. The microdata include all monthly bills to each sampled household, and further state which bills are based on actual readings and which are based on estimates. Since my purpose is to examine actual use, I integrate billed use over each period elapsed from one observed actual reading to the next. I express use as a 30-day rate, i.e., in terms of a monthly equivalent in m$^3$ mo$^{-1}$ or kWh mo$^{-1}$.

For example, for a given household $i$, electricity billed in March, April, and May 2015 was based, respectively, on actual (A, 350 kWh), estimated (E, 370 kWh), and actual (A, 390 kWh) meter readings on March 5, April 5, and May 5. I then observe actual use of $370 + 390 = 760$ kWh over the 61-day period $t$ that starts on March 6, 2015 and ends on May 5, 2015. Electricity use is $760/61 \times 30 \approx 374$ kWh per 30 days. The estimation sample (for electricity) would include one observation of 374 kWh mo$^{-1}$ for the bimonthly period March 6 to May 5, 2015. Since SP Services staff distribute their visits evenly across residences, usage periods vary across households; for example, the 61 days closing on May 1 for some households, the 61 days up to May 2 for others, and so forth (Supplementary Fig. 10). This adds to sample variation in heat exposure[14]. E-mail correspondence with SP Services informed that longer periods than 2 months happen due to a utility meter being inside the customer's premises where the meter reader cannot access when visiting or the meter is faulty. In such cases, an investigation takes place and the faulty meter is replaced. But until the replacement is complete, the readings can only be estimated. In such cases, I integrate over several monthly bills to the next meter reading. Shorter periods than 2 months are due to (relatively few) customers calling in with actual readings within 1 month of a meter reader's visit, in time for the next bill. Supplementary Fig. 11 shows the distributions of duration for water and electricity (actual) use observations. Supplementary Fig. 3 reports the distributions of the number of periods of water use and electricity use observations across households.

The dependent variable in regression model (1) is the natural logarithm of resource use in monthly equivalents ($m^3\,mo^{-1}$ or $kWh\,mo^{-1}$). Concurrent ambient temperature over the usage period, $T_t$, is the key variable of interest, with parameters $\lambda_i$. I allow the impact of temperature on resource use to vary by apartment type, by including interactions between $T_t$ (or a nonlinear function of $T_t$) and a set of indicators for the different apartment types. In a linear specification, justified by the narrow tropical temperature range, an apartment type-specific temperature effect is denoted $T_t\lambda_i$, i.e., $T_t\lambda_{1\text{-}room}$ if the household lives in a 1-room apartment, $T_t\lambda_{2\text{-}room}$ in a 2-room apartment, and so on. A linear specification would not be applicable if one were inferring responses to North American winter and summer temperatures in the same regression model[29].

Other potential environmental drivers of water and electricity use, also integrated over each actual usage period and denoted by $X_{it}$, include relative humidity, dew point depression, wind speed, the incidence of rain, and $PM_{2.5}$. Where multiple weather stations measure the same variable (temperature, wind, and rain), I average across locations, given the tight spatial correlation in weather across the city–state. I assign $PM_{2.5}$, observed separately for Singapore's north, east, south, west, and central districts (Supplementary Table 3), based on a household's postal code in the microdata (there are 73 such geographic areas).

Household fixed effects, $\phi_i$, capture unobserved heterogeneity due to family composition and tastes, as well as device and building characteristics. These account for all averaged-over-time shifters that vary by user-dwelling identifier, including number, age, and occupation of members; household income and wealth; conservation attitudes; and adoption and resource efficiency of appliances.

Vector $\delta_t$ includes different time-varying controls. First, month fixed effects, based on the last day of period $t$, account for seasonality. Second, year fixed effects (similarly, a time trend) capture economic growth. Third, $\delta_t$ includes the proportion of days in period $t$ falling on different days of the week, public holidays (when households may be at home), and school holidays (when households may travel). I can allow time controls to vary in the cross-section, $\delta_{it}$; for example, by specifying interactions between apartment-type indicators and year fixed effects, accounting for economic shocks that vary over the socioeconomic distribution. Similarly, families with children may live in bigger apartments and I allow interactions between apartment-type indicators and the proportion of school holidays. In terms of climate, it is worth noting that there is significantly less seasonal variation in the tropics compared to a temperate climate. While this feature limits the range of temperature over which households respond—for example, we do not learn about impacts of winter temperatures in Europe—it provides a natural control for the key climate component of seasonal variation. Estimates are similar if month fixed effects are based on the midpoint of period $t$ rather than its last day, and if I specify (52) week fixed effects rather than (12) month fixed effects.

An estimation sample consists of household by usage period observations from September 2012 to December 2015 in the full apartment sample trimmed at percentiles 1 and 99 of water or electricity use by apartment type. Estimates are similar without trimming the estimation samples, and instead implementing regressions on the full water or electricity use sample.

Econometric residual $\varepsilon_{it}$ is an unobserved shock to household water or electricity use. Model (1) is estimated by OLS. The identifying assumption is that, conditional on controls, the residual is uncorrelated with $T$; in particular, $E[T_t\varepsilon_{it}|X_{it},\phi_i,\delta_t]=0$. Water prices have not changed, and in a robustness test I add downward-trending electricity prices in the electricity equation rather than relying on year fixed effects.

**Inclusion and plausible endogeneity of $PM_{2.5}$.** Singapore experiences variable levels of air pollution from both onshore and transboundary sources, including very high $PM_{2.5}$ in mid-2013 and late 2015 due to land fires in equatorial Asia (Fig. 2d), and influenced by atmospheric ventilation conditions. Due to defensive behavior, pollution may be a driver of household resource demand. For example, to avoid exposure to poor air quality, households may stay more at home, flush the toilet and eat meals at home more often, or shut the windows, turning on the AC. Thus, $PM_{2.5}$ is a control in the household resource demand Eq. (1).

A concern that may arise when adding $PM_{2.5}$ as a regressor is reverse causality: electricity or water demand shocks $\varepsilon$, unobserved by the empirical researcher, may lead to higher emissions and pollution—e.g., from electricity generation to meet electricity demand, or economic activity that shifts both resource demand and $PM_{2.5}$ via emissions. On top of this, $PM_{2.5}$ may correlate with temperature, the main variable of interest. In addition to OLS, which restricts regressors to be orthogonal (conditionally exogenous) to unobserved demand shocks, I present results based on an alternative estimator that allows $PM_{2.5}$ to be endogenous. The 2SLS estimator I specify assumes that a household's resource use responds to shifts in regional land fires and shifts in atmospheric ventilation/stagnation only indirectly, through these variables' impact on ambient $PM_{2.5}$. These instrumental variables are assumed to be orthogonal to unobserved demand shocks. Intuitively, co-variation between household demand and the instruments works through pollution to demand, producing unbiased estimates of the impact of pollution and its correlates such as temperature.

Consider two sets of instrumental variables that are excluded from the household resource demand equation and are thus uncorrelated with omitted drivers of household demand $\varepsilon$: The first set, denoted by $F$, captures land fires spatially aggregated over Southeast Asia[72]. the second set of instruments, denoted

by $A$, captures atmospheric conditions in Singapore, namely, temperature-altitude gradients and wind direction[73]. This exogenous component of $PM$, denoted $\widehat{PM}$, is fitted in a first-stage regression:

$$PM_{it} = F'_t\Delta^F + A'_t\Delta^A + W'_t\Delta^W + \phi_i + \delta_t + \epsilon_{it} \qquad (2)$$

[first stage of 2SLS]

In the first-stage regression, pollution shifts with season and exogenous weather $W_t$, such as surface temperature, wind speed, and precipitation, included in demand model (1). Pollution further shifts with instruments $F_t$ and $A_t$, which households do not respond to directly. The exclusion restrictions are that a household's water use and electricity use respond both to shifts in land fires (close and upwind vs. far or downwind), and to shifts in atmospheric dissipation (a stagnant vs. ventilated atmosphere) only through these instruments' influence on $PM_{2.5}$. Supplementary Table 3 describes the excluded instruments (including sources). The 2SLS estimator yields estimates of temperature's impact on utility demand that are very similar to OLS.

**Testing for measurement error in temperature.** To check for possible attenuation bias from classical measurement error in the main variable of interest —temperature, as measured by Meteorological Service Singapore (MSS)—I perform the following robustness test. I add $T_t^{MSS}$ (temperature measured by MSS) to the vector of endogenous variables, and add temperature as measured by an independent source, $T_t^{NUSG}$, to the vector of instruments. The superscript denotes the Kent Ridge weather station maintained by the National University of Singapore's Department of Geography. While NUSG's site on the vegetated and less urban Kent Ridge campus is slightly cooler than the average MSS weather station (Supplementary Table 3), the independently measured temperature series are very tightly correlated, suggesting that attenuation bias is not present, as indeed I find.

**Linearity of temperature effects.** In view of the narrow temperature range in tropical Singapore, the baseline specification imposes some form of linearity in the resource demand response to temperature. The demand response is assumed to be linear to daily mean temperature averaged over the usage period (Fig. 3, empirical support 26.1–29.1 °C), or to the proportion of days in the usage period with daily mean temperature above 28.5 °C (Fig. 4, empirical support 0–83%).

As an alternative to a linear response, I specify average temperature entering the demand equation via highly granular bins of width 0.4 °C, in particular indicators for average temperature during the usage period in the following ranges in °C: [26.9–27.3], [27.3–27.7], [27.7–28.1], [28.1–28.5], and ≥28.5 (the reference category is <26.9 °C). Supplementary Fig. 4 shows a precisely estimated linear electricity demand response over the empirically observed temperature range. The estimated water demand response, while increasing, is less clearly linear and less precise. As in Fig. 3/Table 1, standard errors on the estimated temperature effects on water use are about 50% larger than on electricity use.

**Heat-relief behaviors stated by Singapore households.** The article's main findings are based on observational evidence; specifically, household behavior as revealed by resource consumption choices. To complement such evidence, and motivated by a reviewer, I conducted a short survey in September 2018 on the Qualtrics online platform to investigate the convenience or prevalence of showering, washing and laundry as water-based heat-relief strategies. After a quality check (Question 1), respondents were primed with a picture of a person sweating under the sun, asked to consider "a very hot weekday in Singapore this September" in which their "daily routine has been normal…," and asked to select at most three heat-relief strategies that were more likely to apply to their home (Question 2). Alternatives included use of the AC, electric fan, shower (more often or for longer), washing one's face, putting wet towels in the refrigerator to place on the body once cool, spending more time at home, and laundry, besides "other" and "none of the above." Supplementary Table 7 provides the exact wording. I did not include water (swamp) coolers, as these do not seem prevalent—at least for indoor use—in the humid tropics. Indeed, no respondent selected "other" and entered "air cooler" or the like. I randomized the order in which heat-relief alternatives appeared by respondent, except "other" and "none of the above" which were shown at the end. The final question of the survey (Question 3) enquired about the respondent's age, gender, position in the household, dwelling type, and home appliance portfolio.

I targeted 300 respondents and initially specified representative distributions for gender and age, trimmed at 20 years from below and 85 years from above. I restricted households to those living in apartments, leaving out the 6% of house dwellers in the population. I initially chose similar quotas across apartment types. I subsequently relaxed the quotas in order to reach 311 valid responses (six responses were invalidated, e.g., one respondent stated to be a 30-year-old parent of the household head). In particular, there were relatively few respondents aged 65+ or living in 1- or 2-room apartments. The first question comprising a quality check followed Qualtrics' best practice, in which a respondent needed to select "I will provide my best answers." To measure a respondent's attentiveness, I tracked the time between clicks. Most respondents indeed selected no more than three alternatives, as instructed, in the second question.

**Code availability**. All computer code used to generate published results can be accessed at https://goo.gl/SdxJ3Y.

## Data availability

Proprietary household-level utility usage microdata can be purchased from SP Services Ltd. To allow approximate verification, water and electricity use data aggregated to usage period by two-digit postal code by apartment type triple can be accessed at https://goo.gl/SdxJ3Y. Moreover, upon reasonable request, the microdata are available on an NUS Department of Economics (or equivalent institutional) computer to replicate all published results from the deposited computer code. All other data can be accessed at https://goo.gl/SdxJ3Y, including the source files used to prepare Figs. 1, 2, and 5b, Supplementary Figs. 1 and 2, and Supplementary Tables 2 (except utility microdata), 3, and 7.

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

## Acknowledgments

I am grateful to Phyllis Ang, Jeanne Cheng, Teck Hua Ho, Ivan Png, Benjamin Soo, Lionel Wee, Melvin Wong, and Julian Wright for facilitating access to SP Services microdata, as well as the organizations mentioned in the text for sharing other data. I thank Eric Fesselmeyer for volunteering household income from the Household Inter-view Travel Survey. I thank Fu Ginn Cheong and Yue Feng Toh for research assistance and acknowledge support from Singapore's Ministry of Education Academic Research Fund Tier 1 (R-122-000-235-112).

## Author contributions

The author is responsible for all parts of the research.

## Additional information

**Competing interests:** The author declares no competing interests.

