## [Peer Review File · Nature Communications]

Reviewers' comments:

Reviewer #1 (Remarks to the Author):

GENERAL COMMENTS

I wonder about the general thermodynamic efficiency of evaporative cooling strategies within a climate with such high ambient humidity levels. The linkage between the presence of refrigeration appliance adoption and water based cooling strategies is brushed aside by the author, despite the high penetration levels of refrigeration appliances among the population overall (93%). It seems that in the absence of A/C, refrigeration units would provide an obvious mechanism for achieving cooling without the need for evaporation. For example, using a refrigerator to cool wet towels which are then placed on the body, or for the production of ice. This could be an important alternative strategy among intermediate income households (dwellings without A/C but with refrigeration) that has not been considered in the analysis - one which conflates the consumption of electricity with the consumption of water for cooling services.

Throughout the paper the author refers to household income level and dwelling size as being almost interchangeable. While these two variables are certain to be strongly correlated, this correlation will not be perfect. Thus, the language used to describe the methods should more transparently reflect the fact that, separate from the micro-data case study, income was not directly observed as an independent variable, only housing characteristics were.

The core finding of differential increase/decrease in water/energy consumption with temperature variation is solidly supported by the analysis, though not unexpected or un-intuitive (at least for this reviewer). This finding needs to be placed within the context of the broader trajectory of the regional energy/water systems to help the reader determine if it is actually a problem or something that just "is."

Overall, the conclusions of this study leave much to be desired. For example, what is the significance of the observation of these differential strategies for cooling within the context of a changing climate? Is Singapore expected to experience higher heat in the future? What about its prospects for increased economic development, and thus higher A/C penetration and utilization levels going forward? Are these outcomes that should be avoided through the implementation of some set of

policies that would force high income residents to adopt the more energy efficient (but more water intensive) cooling strategies of lower income residents? How likely is such an approach to be feasible? I am not a regional expert, but my understanding is that Singapore is highly freshwater constrained. Would the pursuit of a more water intensive set of residential cooling strategies actually be detrimental given these local challenges?

SPECIFIC COMMENTS AND SUGGESTED REVISIONS

Page1 / Para1:

[COMMENT]

Asia and the Mediterranean certainly aren't the only regions with cities experiencing reduced water availability. Should expand the scope of this statement to encompass cities globally (with appropriate supporting references).

Page1 / Para1:

[REVISION]

"price {based} and non-price {based}"

Page1 / Para2:

[REVISION]

delete "To my knowledge"

[COMMENT]

If this pronouncement of fact is being made it should be based upon the findings of the literature survey. The following reference discusses a case study investigation which mentions (without explicit

quantification) the interaction of water consumption with air conditioning demand in a rural area of Iran.

REF:

A.R. Keshavarzi, M. Sharifzadeh, A.A. Kamgar Haghighi, S. Amin, Sh. Keshtkar, A. Bamdad,
Rural domestic water consumption behavior: A case study in Ramjerd area, Fars province, I.R. Iran,
Water Research,
Volume 40, Issue 6,
2006,
Pages 1173-1178,
ISSN 0043-1354,
<https://doi.org/10.1016/j.watres.2006.01.021>.

(<http://www.sciencedirect.com/science/article/pii/S0043135406000406>)

Keywords: Domestic water use; Rural households; Rural water consumption; Ramjerd; Iran

Additionally, the reference below also provides a useful review of previous literature relating drivers of household water demand.

REF:

C. Fox, B.S. McIntosh, P. Jeffrey,
Classifying households for water demand forecasting using physical property characteristics,
Land Use Policy,
Volume 26, Issue 3,
2009,
Pages 558-568,
ISSN 0264-8377,

<https://doi.org/10.1016/j.landusepol.2008.08.004>.

(<http://www.sciencedirect.com/science/article/pii/S0264837708000884>)

Keywords: Household water demand; Water supply; Demand forecasting; Housing development; Planning

Page1 / Para3:

[REVISION]

"from whom I {have obtained} my data"

Page2 / Para1:

[REVISION]

delete "informative"

Page2 / Para2:

[COMMENT]

The author is co-mingling the results of their own study with the discussion of the background issues and introduction of literature results. Results from this study should be withheld until later sections, after the methods have been introduced and described.

Page2 / Para5:

[REVISION]

Run-on sentence at "Consistent with..." Should be split up for clarity.

Page3 / Para1:

[COMMENT]

The comparison of relative amounts of total spending - on water vs. electricity - for households at different ends of the wealth distribution is only valid if the tariff structures for each of the two commodities are equivalent. In many areas of the world, electricity tariffs can be quite complicated, with demand charges, time of use rates, baseline allocations, etc. The author needs to discuss these factors to help the reader with their interpretation of the significance of the findings regarding the differences in the total expenditures between different groups.

Page3 / Para2:

[COMMENT]

So for clarity: account numbers correspond to the unique combination of user/dwelling. Is it possible to tract dwellings and/or users independently using the information available from this provider?

Page3 / Para2:

[COMMENT]

How is the restriction of the data to apartment units applied? Is it based upon information contained within the address - i.e. presence of unit/apt decorators - or does the dataset provide additional building attribute information for the dwellings involved?

Page3 / Para3:

[REVISION]

Sentence beginning with "How Singaporeans!..." is confusing and poorly structured. Needs revision.

Page4 / Para1:

[COMMENT]

The author references a US national ambient air quality standard. Is this because no similar standard exists locally/regionally?

Page4 / Para2:

[REVISION]

The literature does not study. Revise this sentence for clarity.

Page4 / Para2:

[COMMENT]

Information about price schedules included in this section should be revealed earlier, per the previous comment.

Page5 / Para3:

[COMMENT] The author continually refers to income distribution, when they are actually describing housing size distribution. Though the two are likely strongly correlated, we don't know exactly what the strength of that correlation is.

Page5 / Para4:

[COMMENT]

The characteristics of the housing stock in Santa Barbara, CA are likely to be extremely different from those in Singapore. This, I suspect, will hold particularly true with regards to the characteristics of high density multi-family residential units - which I understand dominate, in Singapore - though are virtually non-existent in Santa Barbara. Perhaps, another comparative example could be found.

Page6 / Para3:

[COMMENT]

Undefined Acronym (2SLS). Indicate that this refers to two-stage least squares.

Page7 / Para4:

[COMMENT]

Always indicate the base of the logarithm used (i.e. base 10, natural log, etc.)

Page7 / Para5:

[COMMENT]

This choice of temperature threshold as defining high heat seems a bit odd. In imperial units 95 deg. fahrenheit is typically used. I know that in SI units 30 degrees Celsius is typically used. Perhaps explain the reasoning behind this choice a bit?

Page8 / Para3:

[COMMENT]

A linear of electricity and/or water consumption to temperature variation is likely going to be a very strong assumption - and perhaps an overly conservative one. Observations from other research suggest that these responses, particularly on the energy side, may not be linear.

REF:

Morna Isaac, Detlef P. van Vuuren,

Modeling global residential sector energy demand for heating and air conditioning in the context of climate change,

Energy Policy,

Volume 37, Issue 2,

2009,

Pages 507-521,

ISSN 0301-4215,

<https://doi.org/10.1016/j.enpol.2008.09.051>.

(<http://www.sciencedirect.com/science/article/pii/S0301421508005168>)

Keywords: Residential energy demand; Cooling; Heating

Page8 / Para5:

[COMMENT]

The fact that the increases and decreases relative to the socio-economic distribution are monotonic is a direct consequence of the author's assumptions of linear variation. This is not a finding therefore, so much as it is a direct consequence of the modeling framework as it has been constructed.

Page9 / Para3:

[COMMENT]

The following sentence seems to be tautological with respect to the study's research question:

"There is a tradition of using residential durable goods portfolios as correlates of income, and since many durable goods are operated with electricity, a market's average electricity consumption may reasonably proxy for its average income level (Fig. 1A)."

Here, it seems that you are using electricity consumption levels as a proxy for income levels. Then, immediately after, using this proxy as evidence of relationships between electricity consumption at different income levels. I am a bit concerned by the validity of this approach.

Perhaps these conclusions should have been set aside for the analysis performed with the HITS micro-data?

Page11 / Para2:

[REVISION]

Awkward phrasing at "the channels of behavior..." consider revision.

Page11 / Para2:

[REVISION]

Consider revising: "The data are limited in the sense that they do not detail consumption by {end-use category}."

Reviewer #2 (Remarks to the Author):

I have attached my comments to the author as a pdf file. The tex code that generate this is:

%This file will create a simple one page document.

```
\documentclass[11pt]{report}
```

```
\setlength{\evensidemargin}{0 in} \setlength{\oddsidemargin}{-0
```

```
in} \setlength{\textwidth}{6.5in} \setlength{\textheight}{9.0in}
```

```
\setlength{\topmargin}{-0.5in} \linespread{1.2}
```

```
\begin{document}
```

```
\noindent \large \textbf{Manuscript Title:} Electrical Appliances Moderate Households' Water Demand Response to Heat\textbf{Journal:} Nature - MS\# 2017-01-00172B\textbf{Date of Comments:} July 31, 2018\vspace{0.15in}\hrule\vspace{0.15in}\noindent\textbf{Summary: } This paper uses a novel household level dataset on water and electricity consumption for Singapore to estimate the response of residential water and electricity consumed to temperature. This paper adds to a nascent literature, which examines consumption patterns in a world with a changing climate. The existing literature has focused largely on Europe and North America. Temperature dependent consumption responses are not well understood in low and medium income countries. Singapore is wealthy by global standards, yet has significant income heterogeneity within the
```

country. Its climate is hot and air conditioning is standard for richer, yet not poorer households. It in some sense is a perfect location to study a 'transition' - using within - not across country variation. Further, while there is much talk about a 'water energy nexus', few papers have empirically explored the relationship between households' electricity and water consumption. This paper does nice job doing just that, all the while applying state of the art statistical techniques from the causal inference literature. The author finds that low income households increase their water consumption during periods of heat, while no such response is detected for high income households. Further, only high income households display increasing consumption of electricity during heat events. This is consistent with a classical model of durable goods adoption, where wealthier households are more likely to adopt pricey durables such as refrigerators and air conditioners. \\\hline

\hline

\vspace{0.15in}\noindent\textbf{My Comments:}\begin{enumerate}\item The paper's main point is that higher income households increase their electricity consumption by more when it is warmer outside, compared to poorer households. The reverse is true for water consumption. The author does not observe income, but from aggregate statistics shows that bigger apartments are owned by richer people. This is not surprising. The author also shows that bigger apartments have more people in them on aggregate. He does not observe the number of people per household for the data used in estimation. I have read this paper four times to look for clarification on this, but I think the dependent variable is household electricity consumption (no per capita electricity consumption). This means that the results should be interpreted as 'richer and bigger' households have a stringer electricity response to heat and a smaller water response to heat. The water results are still really interesting in this case. It means that smaller households shower disproportionately more than bigger and richer households. The electricity heterogeneity is a little bit less interesting through this, I think. Or convince me of the opposite. Bigger apartments with more people in them physically require more electricity to get cooling services per capita because of the increased volume of air that needs to be cooled. I am seeking a much stronger explanation of the impact of the 'per capita' versus 'higher income' effect here.

\item I would like to also suggest thinking about the interpretation of the results. While I believe the empirical findings of the paper - namely that poor people increase their water consumption in response to heat events, I wonder if this is in some sense a 'convergence' response. If rich people find water relatively inexpensive and 'shower at will' while poor people are more hesitant due to the cost, maybe both have the same 'water consumption bliss point'. Maybe two showers a day is the bliss point when it's hot. Poor people do not consume that much but rich people do. What this means is that as incomes rise, poor people consume water up to that bliss point and then install air conditioners and in addition to showering more (up to the bliss point), consume air conditioning services. This suggests a consumption ladder in some sense, which is similar to what the appliance

adoption literature has observed. Table S1 in the supplemental materials sort of gets at this. I am not sure you can test for this, as you do not know how many people live in the bigger apartments and hence income is confounded by number of occupants. But it would be important to talk about what this means in the larger economics/income literature. Can we expect water consumption to plateau and electricity consumption to grow nonlinearly (in temperature)? In the introduction you talk about them almost as if they were substitutes, which I am not sure is the only/right interpretation.

\item How does your temperature response for wealthy household's electricity consumption compare to what others have found in the North American and European context? The response should be in the same ballpark. It would add credibility to the study to point this out.

\item This is the first paper that actually gets at the water - energy nexus in a way that is actually interesting in my view. I would think about pitching this around that mystical concept. At least in the introduction.

\item In the first paragraph I would make it clearer that there is a quantity and quality issue arising from climate change - both for water and electricity (if we think of reliability as a quality issue).

\item On your page 2, you claim that showers and washing machines are more widespread than ACs. You should cite something from the appliance adoption literature that shows this to be true.

\item In maybe a follow up project, can you identify movers and use those as a source of identification? If not, you should make that clear in the manuscript.

\item Page 5 reports that mean household...

\item I was puzzled by the $\$/m^3/kWh$ measure, but really like it now!

\item Stick with a single currency in the paper.

\item Why is the last day of the bill the right assignment of month for your measure of seasonality?

\item Why is clustering by household right? Your unit of treatment is billing period and weather station? Justify your standard errors.

\item The table and figure notes are super excessive. The tables are also too big. I would edit these down and move what I could to the SI.

\end{enumerate}

\end{document}

Reviewer #3 (Remarks to the Author):

The paper analyses the dependence of water and electricity demand on climatic factors, keeping into account the income (included as a proxy, through information on the apartment-type).

It is a very interesting subject, extremely novel since not yet explored, to my knowledge in the literature and of extreme importance in terms of both real-time forecasting of water and energy consumptions (for the multi-utilities) and of future scenarios of customers' needs, for policy-makers.

On the other hand, the paper is pretty repetitive: from the abstract on, the key result (that the increase in temperature makes the rich households to increase air-conditioning and makes the poorer families increase the water use) is repeated too many times.

In addition, the results are probably more specific than claimed, since the situation of Singapore, in terms of the combination climate + income, is probably pretty unique.

Another issue that should be clarified is why the author assumes (at least initially since eventually the results do not conform it) that there should be an influence between air pollution and water use: I confess that the link is not apparent to me...

Most importantly, the paper should better describe the available data sets (this is crucial when applying any data-driven methodology) and also the applied statistical techniques, in order to better justify the conclusions that are drawn. The information on the data is scattered in the different sections in a not orderly way, often AFTER having presented some figures based on them or some results.

And both in the Table description at p. 8 and in Section V: the discussion of the results are pretty heavy to follow, with too many numbers reported in the text: a few illustrative figures should make the paper much shorter and easier to follow (after all, the scientific content, even if interesting, is not so much...).

I would suggest to summarize the results into a much shorter technical note, with a clearer description of the data and the methodology and a more visual, friendly description of the results.

DETAILED COMMENTS

p. 2, l. 8: do not refer to “log-points” but to the values in their actual unit and the percentage of increase/decrease.

p. 2, ll. 5-11: there is too much detail for an introduction section, too much of the results is anticipated in this paragraph in quantitative terms, that should instead go only into the Discussion of the results section.

p. 2 (beginning of Section 2) and Fig. 1A: Please explain which data were used for obtaining the numbers shown the figure? Expenditure data are not provided by the utility of course.

p. 3, l. 3-5: Is water tariff dependent on the monthly volumes? This is partly explained at p. 4, but it should be here.

p. 3: explain here (and not at p. 4 & 5) which other information is available on the costumers (apartment size and value, number of members in the family,...).

p. 4: there is information on water conservation measures and tariffs but not on energy conservation tariffs and energy tariffs.

p. 5: Table S.2 and Fig. 3: where does this information come from? The source is not clear.

In addition, the number of members of the family is a very important information too and it seems to be missing...

p. 5, l. 10: actually how do we know if the apartments are naturally ventilated or not? The assumption of the relationship between apartment size/income/no air conditioning has not been proved or sustained by actual data...

p. 5: the information in terms of percentages is not fully informative: there is the need to provide the results terms of both volumes and percentages (and stating that 25% of the small ('poor') households use 20% of water volumes does not support the conclusion that poor family consume more water... Data reflecting both actual volumes and number of family members are needed.)

p. 5, l 22: DOS? Please put the source in clear and adding the year of the publication. Clarify also which information is available on family size? A mean depending on apartment size?

p. 5 (end): Can you discuss the differences between the results obtained in the present case study and in the Santa Barbara one? (and note that for the case study of Santa Barbara the actual volumes - and not percentages - are correctly reported. ..)

Section II is not clear, more details are needed on the statistical procedures.

Eq. 1: All the terms presented in equation 1 should be defined either before or just following the equation and not too much later on.

Text following Eq 1:

- The phrase on the meter readings is not clear: how do you go from bimonthly to monthly data: assuming the same values for the two months?
- Not clear what $T(t) \cdot \lambda(i)$ means: isn't the apartment typology represented by λ alone?

p. 7, End of Section 3: please define what "2SLS" means.

p. 8: "interact the variable"?? what does it mean?

In this response letter to Reviewer #1, **verbatim edits and additions to the manuscript are highlighted in yellow**. I omit minor edits intended to improve readability and that do not change content.

Reviewers' comments (in italics):

Reviewer #1 (Remarks to the Author):

GENERAL COMMENTS

I wonder about the general thermodynamic efficiency of evaporative cooling strategies within a climate with such high ambient humidity levels. The linkage between the presence of refrigeration appliance adoption and water based cooling strategies is brushed aside by the author, despite the high penetration levels of refrigeration appliances among the population overall (93%). It seems that in the absence of A/C, refrigeration units would provide an obvious mechanism for achieving cooling without the need for evaporation. For example, using a refrigerator to cool wet towels which are then placed on the body, or for the production of ice. This could be an important alternative strategy among intermediate income households (dwellings without A/C but with refrigeration) that has not been considered in the analysis - one which conflates the consumption of electricity with the consumption of water for cooling services.

Thank you for your detailed comments, including those that follow, which I believe have helped me improve the manuscript. With regard to the household behavior described above, such a strategy and others would be subsumed in the water-energy demand responses that my study quantifies. The emphasis on showering, washing and laundry in the discussion of results is due to (i) their end-use share in the home and (ii) their (apparent) convenience or prevalence as a water-based heat relief/remediation strategy.

To support (i), on p.3, I added:

“An observational study by the water agency of end uses in 2016/17 found that showering accounts for 27% of Singapore’s home water use, followed by bathroom tap/basin (18%), flushing (18%), kitchen (16%) and laundry (15%).^{REF}”

Reference added:

PUB. *Showering Tops Water Usage in Household Water Consumption: Singapore Household Water Consumption Study 2016/2017*. Public Utilities Board, Singapore’s National Water Agency (2018).

To investigate (ii), I conducted a short survey on the Qualtrics online platform among 311 Singapore households. The survey showing the exact wording of the questions posed, with separate layouts for desktop and mobile, is attached to this response. After a quality check,¹ respondents were primed with a picture of a person sweating under the sun, asked to consider “a very hot

¹ I followed Qualtrics’ best practice, such as a first question comprising a “quality check,” where a respondent needed to select “I will provide my best answers.” To measure a respondent’s attentiveness, I tracked the time between clicks. Most respondents indeed selected no more than three alternatives, as instructed, in the second question. I specified representative distributions for age (trimmed at 20 years from below and 85 years from above) and gender of the respondent. I restricted households to those living in apartments (leaving out the 6% of house dwellers in the population) and, to increase sampling precision among respondents in 1- and 2-room apartments, I raised the subject quotas for these apartment types relative to their share in the population.

weekday in Singapore this September” in which their “daily routine has been normal...,” and asked to select at most three heat-relief strategies that were more likely apply to their home. Alternatives included use of the AC, electric fan, shower (more often or for longer), washing one’s face, putting wet towels in the refrigerator to place on the body once cool, spending more time at home, and laundry, besides “other” and “none of the above.”² Importantly, the survey randomized the order in which heat-relief alternatives appeared by respondent (except “other” and “none of the above,” which were shown at the end). A third question collected age, gender, position in the household, dwelling type, and appliance portfolio.

The following table, reproduced from panel A of new Table S.7, reports the proportion of respondents selecting each behavior (for brevity I do not reproduce panel B here):

Table S.7. Stated heat-relief behaviors in an own survey of Singapore households. A., Number and proportion of respondents selecting the different alternatives (the order in which these appeared were randomized by respondent, except for “other” and “none”). **B.,** Distribution of demographic characteristics over respondents.

It is a very hot weekday in Singapore this September . Your daily routine has been normal , attending school or work, running errands, going to the community centre, and so on. Which of the following may apply to your home that day or evening? Please tick at most three (one, two or three) statements that are more likely to apply:	Number of respondents (of N=311 in total)	%
Due to the heat, at home we/I shut the windows and turned on the air conditioner , compared to a cooler day.	113	36%
Due to the heat, at home we/I turned on the electric fan , compared to a cooler day.	187	60%
Due to the heat, at home we/I showered more often or for longer , compared to a cooler day.	121	39%
Due to the heat, at home I washed my face more often or for longer , compared to a cooler day.	55	18%
Due to the heat, at home I put wet towels to place on my body once cool , compared to a cooler day.	15	5%
Due to the heat, I spent more time at home , compared to a cooler day.	135	43%
Due to the heat, at home we/I put more dirty/sweaty clothes in the washing machine , compared to a cooler day.	57	18%
Other (open-ended response) _____	1‡	0%
None of the statements above are more likely to apply at home due to the heat, compared to a cooler day.	7	2%

‡ A respondent wrote: “We drink more water each day”

Borrowing from the reviewer’s ideas as well as words, to the Discussion on p.12, I added: “Other cooling behaviors are conceivable. For example, given widespread residential diffusion of refrigerators (Fig. 1A) and relative humidity averaging 76% (Table S.3), one mechanism for achieving cooling without the need for evaporation could be to use the refrigerator to cool wet towels that one then places on the body. A 300-person survey targeted at the general population suggested that the water-based cooling behaviors that are prevalent in Singapore residences take place in the shower (used more often or for longer), wash basin and laundry (Table S.7).”

Throughout the paper the author refers to household income level and dwelling size as being almost interchangeable. While these two variables are certain to be strongly correlated, this

² I decided against the inclusion of water (swamp) coolers in the survey, as these do not seem prevalent—at least for indoor use—in the humid tropics. Indeed, no respondent selected “other” and entered “air cooler” or the like.

correlation will not be perfect. Thus, the language used to describe the methods should more transparently reflect the fact that, separate from the micro-data case study, income was not directly observed as an independent variable, only housing characteristics were.

Please see my response to this general comment under specific comment/suggested revision “Page5 / Para3” below.

The core finding of differential increase/decrease in water/energy consumption with temperature variation is solidly supported by the analysis, though not unexpected or un-intuitive (at least for this reviewer). This finding needs to be placed within the context of the broader trajectory of the regional energy/water systems to help the reader determine if it is actually a problem or something that just "is."

Overall, the conclusions of this study leave much to be desired. For example, what is the significance of the observation of these differential strategies for cooling within the context of a changing climate? Is Singapore expected to experience higher heat in the future? What about its prospects for increased economic development, and thus higher A/C penetration and utilization levels going forward? Are these outcomes that should be avoided through the implementation of some set of policies that would force high income residents to adopt the more energy efficient (but more water intensive) cooling strategies of lower income residents? How likely is such an approach to be feasible? I am not a regional expert, but my understanding is that Singapore is highly freshwater constrained. Would the pursuit of a more water intensive set of residential cooling strategies actually be detrimental given these local challenges?

In response to this set of comments, as well as specific suggestions by the other reviewers, I substantially revised the Discussion. I added further references on why understanding households' heat relief behavior in the context of global warming is important. I followed Reviewer #2's suggestion by adding, on p.14, a paragraph that links my research to the “water energy nexus.” Reviewer # 2 writes: “*This is the first paper that actually gets at the water - energy nexus in a way that is actually interesting in my view. I would think about pitching this around that mystical concept.*” I now argue that the research can help improve real-time demand forecasting for multi-utilities (p.14). Reviewer #3, in his/her generous word, finds this aspect “*of extreme importance*” and “*extremely novel.*”

I agree with a point raised by Reviewer #3 on Singapore's “combination of climate + income.” On p.14, I acknowledge that more studies are called for, using similar methods applied to household-level panels on joint resource demands for differing locations (even to examine Europe's successive heat waves with varying AC access, p.15). My approach favors institutional depth over breadth, helped by its one location, Singapore, which Reviewer #2 describes as “*wealthy by global standards, yet has significant income heterogeneity within the country. Its climate is hot and air conditioning is standard for richer, yet not poorer households. It in some sense is a perfect location to study.*” I added relevant statistics for some other major cities in tropical Asia to argue that what we learn from Singapore's households is relevant to other locations that lag Singapore in terms of economic development, yet share a similar climate. I expanded:

“*To a varying degree, urban water availability shapes urban development, as has been Singapore's case at least over the past half century^{REF}. From 1948 to 2016, annual mean temperature rose at a rate of 0.25 °C per decade, and the city-state has invested heavily on new water sources, including desalination and reuse^{REF}.*”

Observational evidence on the multi-utility demand responses to heat by Singaporeans can shed light on how other populations, in tropical Asia and beyond, will respond as incomes rise and the climate warms. In Southeast Asia, climate models project annual temperature to increase by 1-4 °C and winter rainfall to decrease by 20-30% by 2070^{REF}. Only 8% of the 3 billion people living in the tropics currently have AC, compared to over 90% in the US and in Japan^{REF}. One local policy implication for water-stressed cities that are host to a sizable mass of lower-income, pre-AC adoption households, may be to tilt prices in favor of AC use, inducing a drop in water use. Reduced water demand add to the list of health, productivity and other socioeconomic benefits from AC use documented in the literature^{REF}. For example, AC has been credited with a decline in heat mortality in the US over the past century as well as a lower heat mortality in the US than in India (for comparable climate zones)^{REF}. Even in Europe, the press has attributed less excess deaths during a hot 2018 summer to increased AC access compared to the 2003 heatwave; research should investigate these variables along with water demand^{REF}. At the same time, policies that promote AC need to weigh impacts on the aerosol burden and greenhouse emissions from electricity generation and common refrigerants such as hydrofluorocarbons.”

This reviewer asks: *Are these outcomes that should be avoided through the implementation of some set of policies that would force high income residents to adopt the more energy efficient (but more water intensive) cooling strategies of lower income residents?*

In Reviewer #2’s words, my manuscript “*adds to a nascent literature... (that) has focused largely on Europe and North America*”. I advance the hypothesis that in the absence of AC, water demand responds to heat. In this sense, it is my hope that there will be more research to follow. It is beyond the scope of my study to provide an “integrated assessment” of alternative water-energy utilization policies across space and over time, or a comprehensive “benefit-cost analysis” of AC adoption that would offer general policy prescriptions. As I state on p.15, such an integrated analysis will need to consider a wide range of private and social benefits and costs of water and energy/AC use. My aim is to document a joint response of greater relevance to urban planners in some locations than others. For example, a local policy that seeks to promote energy efficiency may be, through water-intensive cooling, detrimental to water security, a tension that the reviewer’s comment captures. In sum, a future study seeking to prescribe policies on household AC adoption and energy use should take account of the impact on water demand—a water-energy “interconnection” first reported here—in addition to other components already documented in the literature, such as health and productivity.

References added:

Burgess R, Deschenes O, Donaldson D, Greenstone M. Weather, Climate Change and Death in India. Manuscript, London School of Economics (2017).

CCRS. *Singapore’s Second National Climate Change Study – Phase 1*. Centre for Climate Research Singapore, Meteorological Service Singapore (2015).

Economist. Air-conditioners do great good, but at a high environmental cost. *The Economist* August 25, (2018).

Heal G, Park J. Temperature Stress and the Direct Impact of Climate Change: A Review of an Emerging Literature. *Review of Environmental Economics and Policy* 10, 347-362 (2016).

Hsiang SM. Temperatures and Cyclones Strongly Associated with Economic Production in the Caribbean and Central America. *Proceedings of the National Academy of Sciences* 107, 15367-15372 (2010).

Isen A, Rossin-Slater M, Walker R. Relationship between Season of Birth, Temperature Exposure, and Later Life Wellbeing. *Proceedings of the National Academy of Sciences* 114, 13447-13452 (2017).

Yuen B, Kong L. Climate Change and Urban Planning in Southeast Asia. *Surveys and Perspectives Integrating Environment & Society* 2, (2009).

Ziegler A, et al. Increasing Singapore's Resilience to Drought. *Hydrological Processes* 28, 4543-4548 (2014).

SPECIFIC COMMENTS AND SUGGESTED REVISIONS

Page1 / Para1:

[COMMENT]

Asia and the Mediterranean certainly aren't the only regions with cities experiencing reduced water availability. Should expand the scope of this statement to encompass cities globally (with appropriate supporting references).

Thank you for pointing this out. I edited the statement as follows:

“...with many densely populated regions in Asia, the Mediterranean **and elsewhere**...”

Page1 / Para1:

[REVISION]

"price {based} and non-price {based}"

I revised the text as suggested.

Page1 / Para2:

[REVISION]

delete "To my knowledge"

I dropped these words as suggested.

[COMMENT]

If this pronouncement of fact is being made it should be based upon the findings of the literature survey. The following reference discusses a case study investigation which mentions (without explicit quantification) the interaction of water consumption with air conditioning demand in a rural area of Iran.

REF:

A.R. Keshavarzi, M. Sharifzadeh, A.A. Kamgar Haghighi, S. Amin, Sh. Keshtkar, A. Bamdad, Rural domestic water consumption behavior: A case study in Ramjerd area, Fars province, I.R. Iran, Water Research, Volume 40, Issue 6, 2006, Pages 1173-1178, ISSN 0043-1354, <https://doi.org/10.1016/j.watres.2006.01.021>.

(<http://www.sciencedirect.com/science/article/pii/S0043135406000406>)

Keywords: Domestic water use; Rural households; Rural water consumption; Ramjerd; Iran

Keshavarzi et al. (2006) states: “rural households use water for both indoor and outdoor purposes. Indoor water use includes consumption for drinking, cooking, hygiene (bathing, laundry, and cleaning), and miscellaneous purposes such as air conditioners” (p.1174). However, Keshavarzi et al. do not relate water quantities with air conditioning. It is not clear to me how air conditioners use water, unless to humidify indoor air—air may be dry, but the article mentions neither indoor nor outdoor humidity.³ Perhaps the authors meant “air coolers,” which use evaporation to cool ambient air. Table 2, titled “Indoor water consumption patterns” and whose second row is labeled “Air conditioner,” is referred to in the text by way of a demographic attribute: “As shown in Table 2, participant indoor activities were grouped into bathing and having air conditioners. Members of 93.2% (n=492) of the households in this study bathed in their own house and 68.6% (n=362) of homes had air conditioner” (p.1176). Air conditioners are not mentioned elsewhere in the article.

In any case, on p.1 I now cite Keshavarzi et al. (2006) as follows:

“A study of 653 rural households in Iran collected information on water consumption and AC adoption but, while mentioning a possible interaction, does not relate the two variables^{REF}”

Reference added:

Keshavarzi AR, Sharifzadeh M, Kamgar Haghighi AA, Amin S, Keshtkar S, Bamdad A. Rural domestic water consumption behavior: A case study in Ramjerd area, Fars province, I.R. Iran. *Water Research* 40, 1173-1178 (2006).

Additionally, the reference below also provides a useful review of previous literature relating drivers of household water demand.

REF:

C. Fox, B.S. McIntosh, P. Jeffrey, *Classifying households for water demand forecasting using physical property characteristics, Land Use Policy, Volume 26, Issue 3, 2009, Pages 558-568, ISSN 0264-8377, <https://doi.org/10.1016/j.landusepol.2008.08.004>.*

(<http://www.sciencedirect.com/science/article/pii/S0264837708000884>)

Keywords: Household water demand; Water supply; Demand forecasting; Housing development; Planning

Thank you for suggesting this valuable reference, which I now include on p.1, when citing the established literature on the determinants of residential water demand. I further cite this reference on p.4 in the context of gardens being a major driver of water demand by families in houses. In a sample of about 500 dwellings in the town of Stevenage, UK, Fox et al. (2009) find that “the precise quantitative comparison of household water demand shows closest agreement between the figures for flats” (p.566), i.e., dwellings in mostly “purpose built apartment buildings” (p.560), and less agreement for semi-detached and detached houses.

Reference added:

Fox C, McIntosh BS, Jeffrey P. *Classifying Households for Water Demand Forecasting using Physical Property Characteristics. Land Use Policy* 26, 558-568 (2009).

³ Relative humidity in Shiraz, Fars province, varies seasonally between 10% and 40%. This is significantly lower than in Singapore. Source: www.worldweatheronline.com/shiraz-weather-averages/fars/ir.aspx

Page1 / Para3:

[REVISION]

"from whom I {have obtained} my data"

I inserted the suggested words.

Page2 / Para1:

[REVISION]

delete "informative"

I dropped "informative."

Page2 / Para2:

[COMMENT]

The author is co-mingling the results of their own study with the discussion of the background issues and introduction of literature results. Results from this study should be withheld until later sections, after the methods have been introduced and described.

Following the reviewer's suggestion, I deleted this paragraph on own findings. The original paragraph followed the convention in the economics literature of describing one's results in the introduction after discussing the context and literature.

Page2 / Para5:

[REVISION]

Run-on sentence at "Consistent with..." Should be split up for clarity.

Thank you for your careful reading. I split the sentence (and elaborated further), as follows: "Fig. 1B shows that the electricity share of household expenditure falls less steeply as income rises compared with the water share of expenditure. This is consistent with the varying adoption of electricity versus water consuming appliances at different points of the socioeconomic distribution, in particular, the wide variation in adoption for AC compared with showers and washers."

Page3 / Para1:

[COMMENT]

The comparison of relative amounts of total spending - on water vs. electricity - for households at different ends of the wealth distribution is only valid if the tariff structures for each of the two commodities are equivalent. In many areas of the world, electricity tariffs can be quite complicated, with demand charges, time of use rates, baseline allocations, etc. The author needs to discuss these factors to help the reader with their interpretation of the significance of the findings regarding the differences in the total expenditures between different groups.

Thank you for pointing this out. In response, when first referencing Fig. 1C on p.3, I added the statement:

"Residential electricity and water prices, described below and reported over time in Fig. 1C, are essentially constant over quantity and hour of use, and are invariant across households."

Inspired by this comment, and also in response to a similar point raised by Reviewer #3, on p.5 I broke out a long paragraph on the "strengths of the combined environment and residential

indoor water/electricity panel and setting” into two paragraphs. The second paragraph discusses water and electricity tariffs at greater length than previously, and opens with a new sentence:

“The relative simplicity and invariance of prices in my setting is also helpful. A large economics literature estimates the price elasticity of household water and electricity demand^{REF}, but here prices are not the object of interest, rather a variable to be controlled for. In fact, prices have either not changed (water) or trended downward (electricity) (Fig. 1C). All households face the same marginal price schedule, both for water and for electricity. Marginal prices are either constant (electricity) or essentially constant (water) over quantity supplied. Water is charged at an increasing block price, as in Hewitt and Hanemann^{REF}, but only 4% of my sample reaches a second block of 40 m³/month, where marginal price is 20% higher. This second block may matter more to the relatively few residential properties that are excluded from my sample, namely houses and bungalows, due to landscape irrigation.”

Page3 / Para2:

[COMMENT]

So for clarity: account numbers correspond to the unique combination of user/dwelling. Is it possible to tract dwellings and/or users independently using the information available from this provider?

Yes, on the account number corresponding to a unique combination of user and dwelling; and no, on the data not allowing one to track the same dwelling over different users, or the same user over different dwellings. On the latter, my understanding is that the provider would be concerned with privacy. Reviewer #2 also requests more clarity on this point. I added the following sentences on p.3:

“Thus, for clarity, each account number corresponds to a unique combination of user and dwelling. By design (account number), the microdata controls for—but does not track—a moving family over different dwellings.”

Page3 / Para2:

[COMMENT]

How is the restriction of the data to apartment units applied? Is it based upon information contained within the address - i.e. presence of unit/apt decorators - or does the dataset provide additional building attribute information for the dwellings involved?

I am grateful for the opportunity to clarify. The original data contains a string variable “premise type” that directly informs on the dwelling type, without the need to infer this from the address. Specifically, I exclude “Terrace House,” “Semi-Detached House,” “Townhouse” and “Bungalow.” In view of this comment, I added on p.4:

“The microdata contains a variable informing the dwelling type, to which I apply the apartment sample restriction.”

(moved forward per Reviewer #3’s advice) “This variable directly informs on the household’s apartment type, according to six standard labeled categories... (i) 1-room... (vi) condominium...”

Further describing the microdata, at this point I add (these statements had previously been included subsequently in the text):

“Utilities consumed by any common areas in a condominium are billed separately to the condominium manager and do not contaminate my sample. The microdata further informs the household’s two-digit zip code, covering 73 geographic areas across Singapore.”

I provide further detail in the notes to Table S.1 (“Summary statistics...” for the microdata):
“A string variable in the microdata informs apartment type, taking on the “values”: HDB01 = 1-room apartment, HDB02 = 2-room apartment, HDB03 = 3-room apartment, HDB04 = 4-room apartment, HDB05 = 5-room apartment, HDB06 = 6-room apartment, HDBEX = executive apartment, PTEAP = condominium apartment. 6-room apartments are rare (only 51 identifiers in the microdata). The labels “HDB” and “PTE” denote the lead contractor for the development of apartments that are sold to individuals, whether the Housing Development Board or a private condominium developer.”

Page3 / Para3:

[REVISION]

Sentence beginning with "How Singaporeans'..." is confusing and poorly structured. Needs revision.

I revised the sentence to:

“Observational evidence on the multi-utility demand responses to heat by Singaporeans can shed light on how other populations, in tropical Asia and beyond, will respond as incomes rise and the climate warms.”

In response to Reviewer #3’s request to reduce repetition, this statement appears only in the Discussion.

Page4 / Para1:

[COMMENT]

The author references a US national ambient air quality standard. Is this because no similar standard exists locally/regionally?

Yes. There is no local or regional air quality standard. Singapore has “air quality targets,” recommended by an Advisory Committee chaired by the National Environment Agency with representatives from different government and academic authorities. The “Sustainable Singapore Blueprint target” for PM2.5, a target for 2020, stipulates an annual mean of 12 $\mu\text{g}/\text{m}^3$. In response, I added on p.5 (as well as edited Fig. 2’s caption):

“Daily mean PM2.5 (particulate matter up to 2.5 μm in diameter) averages 21 $\mu\text{g}/\text{m}^3$. 24-hour PM2.5 routinely exceeds the US primary one-year average National Ambient Air Quality Standard, and the nation’s own “Sustainable Singapore Blueprint target,” of 12 $\mu\text{g}/\text{m}^3$ (Fig. 2D)^{REF}.”

Reference added:

MEWR. *Our Home, Our Environment, Our Future: Sustainable Singapore Blueprint 2015*. Ministry of the Environment and Water Resources and Ministry of National Development (2014).

Page4 / Para2:

[REVISION]

The literature does not study. Revise this sentence for clarity.

Thank you for the correction. In addition to p.5, I revised a similar expression on p.1.

Page4 / Para2:

[COMMENT]

Information about price schedules included in this section should be revealed earlier, per the previous comment.

Please see above for my response to this combined comment.

Page5 / Para3:

[COMMENT] The author continually refers to income distribution, when they are actually describing housing size distribution. Though the two are likely strongly correlated, we don't know exactly what the strength of that correlation is.

In response to this concern, I increased the use of the term “socioeconomic,” which more broadly comprises wealth and expenditure in addition to income. For example, in the paragraph referenced by the reviewer I replaced “(a household’s apartment type serves as a proxy for) its income level” by “its overall socioeconomic standing.” This is not simply semantics: housing is the main component of wealth in a typical Singaporean family’s portfolio. For example, in the last quarter of 2015, residential property assets accounted for SG\$ 300 billion, compared to SG\$ 177 billion for shares & securities (the second largest component), in the “Household Sector Balance Sheet,” which I now reference. Housing is also an important component of overall expenditure, which includes imputed rent for the 90% of owner-occupiers (discussed on p.3). I added this caveat already to the introduction, on p.2, :

“Household-specific income, wealth or expenditure are not directly observable in the microdata. Thus, when estimating a household’s resource demand response to ambient temperature, I take the household’s dwelling type—that I do observe—as a proxy measure of its socioeconomic standing, noting that housing in Singapore is a key component of household wealth.”

Prompted by the reviewer’s comment, I also realized that the Household Interview Travel Survey (HITS), an auxiliary dataset already used in the original version of the manuscript, collected both stated income and dwelling type, allowing me to *check* the strength of the correlation. I added this and other related statements to the paragraph referenced by the reviewer:

“Moreover, in a nation of owner-occupiers, residential property assets are a widespread component of wealth. Housing accounts for one-half of “household net worth,” and values typically increase over the different apartment types (i) to (vi).^{REF} Similarly, rent—whether paid out or imputed in the case of an owner-occupier—is a key component of household expenditure. Rent increases over apartment types (i) to (vi) in line with their value (property prices reflect the present value of the flow of rent). Finally, a separate Household Interview Travel Survey that I subsequently bring into the analysis reports a clear association between income and dwelling type (Fig. S.10).”

To the Results subsection on p.10 that brings HITS-surveyed income into the analysis, I now reference a new figure (reproduced below), as follows:

“From the HITS I take each combination of geographic area by apartment type—again, a “market”—and compute the proportion of households where at least one member states earning over US\$ 1,984 / month, e.g., a household with two parents and two children where one parent selects the (local currency) SG\$ 2,500-2,999 bin or higher. Across the 63 two-digit zip codes covered in the HITS, the median proportion of households with “over US\$ 1,984” individuals is

zero for 1-room apartments, compared to 0.67 for condominium apartments. For each apartment type, Fig. S.10 reports other quantiles across two-digit zip codes for the proportion of households with such high earners; income and apartment type show a clear correlation.”

In the abstract, I state that apartment type is a “verifiable” proxy for socioeconomic status.

Last, but not least, while I do not observe household income in the microdata, the research design *does* very flexibly account for the relationship between resource demands and the household’s income level (and any other demographic and dwelling characteristic), by virtue of the household-specific intercepts. I mention this on p.2:

“It is important to appreciate that the design controls for all unobserved individual household (and dwelling) characteristics, including time-averaged income, besides seasonality and other environmental factors.”

And again in Methods:

“(Household fixed effects) account for all averaged-over-time shifters that vary by user-dwelling identifier, including number, age and occupation of members, household income and wealth, conservation attitudes, and adoption and resource efficiency of appliances.”

References added:

DOS. *Household Sector Balance Sheet (M700981)*. Department of Statistics, Singapore (2018).

Agarwal S, Qian W. Access to Home Equity and Consumption: Evidence from a Policy Experiment. *Rev Econ Stat* 99, 40-52 (2017).

Phang S-Y. Housing Policy, Wealth Formation and the Singapore Economy. *Housing Policy* 16, 443-459 (2001).

New figure, reproduced for convenience:

Fig. S.10. Income and apartment type: Proportion of households where at least one member states earning over US\$ 1,984 / month, by apartment type. An observation is a geographic area of Singapore, specifically, the 63 residential two-digit zip codes covered in the HITS. The box plot shows the median and interquartile range (25th percentile to 75th percentile in the thick bands) in the distribution over geographic areas, by apartment type, for the proportion of households where one or more members stated monthly income of at least SG\$ 2,500 (or US\$ 1,984). Source: Land Transport Authority’s 2008 Household Interview Travel Survey (HITS).

Page5 / Para4:

[COMMENT]

The characteristics of the housing stock in Santa Barbara, CA are likely to extremely different from those in Singapore. This, I suspect, will hold particularly true with regards to the characteristics of high density multi-family residential units - which I understand dominate, in Singapore - though are virtually non-existent in Santa Barbara. Perhaps, another comparative example could be found.

In response to this comment, I added the following statement (the referenced studies are cited in the Fox et al. reference suggested above, for which I am grateful):

“Average water demand by Korean households in 1982, of 17 m³/month in 43 m² flats and 26 m³/month in 105 m² flats, is higher than present-day use in Singapore’s similarly sized 2-room and 5-room apartments, respectively 13 and 19 m³/month^{REF}. More recent consumption in 2000 for flats and apartment units (mean occupancy 1.4 persons) in Adelaide City, a medium density area of Adelaide, Australia, averaged 6 m³/month, compared with 11 m³/month in Singapore’s 1-2 room apartments (occupancy 2.1 persons).^{REF}”

I kept the Santa Barbara comparison as it relates to high-density houses and collected data on household income and water demand, finding no significant association. Moreover, that study was able to collect many other household variables (on a small sample), but did not survey electrical appliances or electricity use.

References added:

Bradley RM. Forecasting Domestic Water Use in Rapidly Urbanizing Areas in Asia. *Journal of Environmental Engineering* 130, 465-471 (2004).

Troy P, Holloway D. The Use of Residential Water Consumption as an Urban Planning Tool: A Pilot Study in Adelaide. *Journal of Environmental Planning and Management* 47, 97-114 (2004).

Page6 / Para3:

[COMMENT]

Undefined Acronym (2SLS). Indicate that this refers to two-stage least squares.

I now refer to two-stage least squares (2SLS) and ordinary least squares (OLS) on first mention, both in the Methods section (after Discussion) and in Results.

Page7 / Para4:

[COMMENT]

Always indicate the base of the logarithm used (i.e. base 10, natural log, etc.)

I now indicate the base of the logarithm (natural). Relatedly, in response to Reviewer #3, I now report results in percent increases in resource use rather than changes in log points.

Page7 / Para5:

[COMMENT]

This choice of temperature threshold as defining high heat seems a bit odd. In imperial units 95 deg. fahrenheit is typically used. I know that in SI units 30 degrees Celsius is typically used. Perhaps explain the reasoning behind this choice a bit?

Thank you for this suggestion. To clarify, this point relates to an alternative measure of heat than average temperature (Fig. 3/Table 1), based on the proportion of days over the usage period in which the daily mean temperature exceeded 28.5 °C (Fig. 4/Table S.4). Further by way of clarification, “above 28.5 °C days” in the sample have a *maximum* temperature, on average, of 33.0 °C.

I now avoid referring to days with daily mean temperature above 28.5 °C as “hot”. In addition to the proportion of “above 28.5 °C days” as a measure of heat, a new Table S.4 provides additional sensitivity analysis showing estimates for specifications that use “above 29 °C days.” Only half as many days (12% of days in the sample) exceed a 29 °C threshold compared to “above 28.5 °C days” (25% of the sample). With a tropical climate (Fig. 1, Table S.3), Singapore did not experience “above 30.5 °C days” (daily mean) during the study period. I added on p.8:

“Table S.4 provides sensitivity analysis to taking the proportion of “above 29.5 °C days” as a measure of hotter usage periods. Only 12% of days in the study period exceed this threshold. Given Singapore’s tropical weather, the daily mean temperature did not exceed 30.5 °C in the study period (Table S.3).”

I also note that a review article that I cite in another context (Heal and Park, 2016) states: “Across a range of contexts (health, labor supply, and labor productivity), micro- and macroeconomic studies find evidence for nontrivial impacts of heat stress, in particular on hot days with temperatures above 78 °F (25 °C)” (p.1)

Page8 / Para3:

[COMMENT]

A linear of electricity and/or water consumption to temperature variation is likely going to be a very strong assumption - and perhaps an overly conservative one. Observations from other research suggest that these responses, particularly on the energy side, may not be linear.

REF:

Morna Isaac, Detlef P. van Vuuren, *Modeling global residential sector energy demand for heating and air conditioning in the context of climate change*, *Energy Policy*, Volume 37, Issue 2, 2009, Pages 507-521, ISSN 0301-4215, <https://doi.org/10.1016/j.enpol.2008.09.051>.

(<http://www.sciencedirect.com/science/article/pii/S0301421508005168>)

Keywords: Residential energy demand; Cooling; Heating

Thank for this valuable reference. I now cite it at different points of the manuscript, for example, when discussing the adoption of residential AC over the socioeconomic distribution in Singapore (Fig. 2 of Isaac and van Vuuren plots the S-curve assumed in their study).

With regard to the linearity of the resource demand response, I clarify that my study focuses on the tropics, where temperature varies in a narrow range. Globally, one can think of nonlinear temperature responses in space (e.g., temperate versus tropical regions) and time (winter versus summer in temperate regions) as “piecewise linear,” with my study examining one piece—the tropics—albeit a vast, densely populated, fast urbanizing and fast growing piece of the world.

Besides linear specifications, I show sensitivity analysis to specifying a nonlinear relationship, for example, on p.8:

“As further sensitivity analysis, Fig. S.5 presents estimates for OLS regression models of water and electricity use with average temperature entering nonlinearly via highly granular bins of width 0.4 °C. The linearity restriction imposed in the specifications of Fig. 3 is broadly borne out by the data.”

I also add the suggested reference on p.15:

“A linear specification would not be applicable if one were inferring responses to North American winter and summer temperatures in the same regression model.^{REF}”

I now tend to refer more specifically to responses to heat than to temperature. For example, the first paragraph in the Discussion states: “increase in water consumption due to a +1 °C **heat shock**.”

Finally, only as a point of comparison, Isen et al. (2017) examine the relationship between in-utero/early-life temperature exposure and later-life earnings for a US population that spans several climate zones. One of their two main specifications uses 10 temperature bins: <0 °C, 0-4 °C, 4-8 °C, ..., 28-32 °C, >32 °C. Besides bins of width 4 °C (other than the corner bins), effects are restricted to be *constant* within a bin. (Their other specification uses five bins: <0 °C, 0-24 °C, 24-28 °C, 28-32 °C, >32 °C.)

Reference added:

Isaac M, van Vuuren DP. Modeling Global Residential Sector Energy Demand for Heating and Air Conditioning in the Context of Climate Change. *Energ Policy* 37, 507-521 (2009).

Isen A, Rossin-Slater M, Walker R. Relationship between Season of Birth, Temperature Exposure, and Later Life Wellbeing. *Proceedings of the National Academy of Sciences* 114, 13447-13452 (2017).

Page8 / Para5:

[COMMENT]

The fact that the increases and decreases relative to the socio-economic distribution are monotonic is a direct consequence of the author's assumptions of linear variation. This is not a finding therefore, so much as it is a direct consequence of the modeling framework as it has been constructed.

This comment is related to the previous one. The specification shown in Fig. 5B restricts the resource demand responses to heat to be linear, but allows the slopes (assumed constant over average temperature) to vary according to apartment type, as a proxy for socioeconomic standing. The monotonicity that the text refers to relates to the empirical finding that the water slopes decrease over the socioeconomic distribution and, in contrast, the electricity slopes increase over the socioeconomic distribution. This is not a construct of the modeling framework. Again, temperature in the tropics varies within a narrow range. Across household by usage period observations in the sample, average daily mean temperature (i.e., averaged over days in the usage period) has a minimum of 26.1 °C and a maximum of 29.1 °C. Fig. S.8 shows very similar results when the dependent variable is resource (water or electricity) demand in levels rather than the natural logarithm of resource demand.

In response, on p.9 I added (bringing forward the discussion of Fig. S.8, panel B):

“Given the narrow range of temperature variation in the tropics compared to a temperate climate such as Europe or North America over the seasons, Fig. 5A’s restriction that resource demand-temperature slopes be constant over temperature is appropriate. Importantly, these slopes are allowed to vary with apartment type. A similar result obtains if I restrict resource demands in levels—rather than their natural logarithm, as in the analysis thus far—to be linear in temperature (panel B, Fig. S.8).”

Page9 / Para3:

[COMMENT]

The following sentence seems to be tautological with respect to the study's research question:

"There is a tradition of using residential durable goods portfolios as correlates of income, and since many durable goods are operated with electricity, a market's average electricity consumption may reasonably proxy for its average income level (Fig. 1A)."

Here, it seems that you are using electricity consumption levels as a proxy for income levels. Then, immediately after, using this proxy as evidence of relationships between electricity consumption at different income levels. I am a bit concerned by the validity of this approach.

Perhaps these conclusions should have been set aside for the analysis performed with the HITS micro-data?

In response to this point, I dropped the statement reproduced by the reviewer, and focused the paragraph on p.9 to document a “lower heat-water response where average electricity demand is higher,” as the now shorter subsection is labeled. Also in response to requests by Reviewer #2 (tables are too long, move what I can to the SI) and Reviewer #3 (less tables), I moved Table 2A, which previously reported results for this subsection, to the supplement. The subsection no longer mentions income, and reads:

“The microdata show substantial variation between the 73 geographic areas in their average electricity demand per household (within apartment type). For 1-room apartments, for example, the distribution of average electricity use across the 73 two-digit zip codes exhibits 5th and 95th percentiles of 109 and 150 m³/month/household. For condominium apartments, 5th and 95th percentiles across the geographic areas are 468 and 894 m³/month/household. This is likely to reflect, at least in part, geographic differences in electrical appliance holdings, such as AC, even within apartment type. To the water demand specification 4 of Fig. 3/Table 1, I include an interaction of temperature with (time-invariant) per household average electricity use for the household’s geographic area and apartment type. Table S.6 documents that the magnitude of the positive heat-water response is lower among households in “markets”—geographic area by apartment type—where per household average electricity use is higher.”

Page11 / Para2:

[REVISION]

Awkward phrasing at "the channels of behavior..." consider revision.

I revised the phrasing to “the heat-relief behaviors.”

Page11 / Para2:

[REVISION]

Consider revising: "The data are limited in the sense that they do not detail consumption by {end-use category}."

Thank you for this suggestion, which I adopted.

In this response letter to Reviewer #2, **verbatim edits and additions to the manuscript are highlighted in yellow**. I omit minor edits intended to improve readability and that do not change content.

Reviewers' comments (in italics):

Reviewer #2 (Remarks to the Author):

Summary: This paper uses a novel household level dataset on water and electricity consumption for Singapore to estimate the response of residential water and electricity consumed to temperature. This paper adds to a nascent literature, which examines consumption patterns in a world with a changing climate. The existing literature has focused largely on Europe and North America. Temperature dependent consumption responses are not well understood in low and medium income countries. Singapore is wealthy by global standards, yet has significant income heterogeneity within the country. Its climate is hot and air conditioning is standard for richer, yet not poorer households. It in some sense is a perfect location to study a "transition" - using within - not across country variation. Further, while there is much talk about a "water energy nexus", few papers have empirically explored the relationship between households' electricity and water consumption. This paper does nice job doing just that, all the while applying state of the art statistical techniques from the causal inference literature. The author finds that low income households increase their water consumption during periods of heat, while no such response is detected for high income households. Further, only high income households display increasing consumption of electricity during heat events. This is consistent with a classical model of durable goods adoption, where wealthier households are more likely to adopt pricey durables such as refrigerators and air conditioners.

Thank you for your encouraging words, some of which I have borrowed to improve the Discussion.

My Comments:

1. The paper's main point is that higher income households increase their electricity consumption by more when it is warmer outside, compared to poorer households. The reverse is true for water consumption. The author does not observe income, but from aggregate statistics shows that bigger apartments are owned by richer people. This is not surprising. The author also shows that bigger apartments have more people in them on aggregate. He does not observe the number of people per household for the data used in estimation. I have read this paper four times to look for clarification on this,

That is correct and is now stated clearly on p.4:

"The microdata does not inform on the individual household's income, size or its adoption of appliances."

but I think the dependent variable is household electricity consumption (no per capita electricity consumption).

The dependent variable is the natural logarithm of household electricity consumption (for the electricity results). The coefficient on temperature thus has the approximate interpretation of a percent increase over the household's baseline use, whether this use is high for a big household or low for a small household. On p.7, I now state (also responding to Reviewer #3):

“Table 1 reports the coefficient on average temperature in regressions of the natural logarithm (log for short) of... The table reports point estimates and standard errors in log points, for a +1 °C variation. Taking the log of resource use as the dependent variable yields temperature coefficients that have the approximate interpretation of a percent increase over the household’s baseline use, whether this use is high for a big household or low for a small household. Fig. 3 presents the same results converted to 95% confidence intervals of percentage increase in resource use (+0.01 log points in use is equivalent to a $e^{0.01} - 1 \cong 1.0\%$ increase, further noting that 95% CI = point estimate ± 1.96 times the standard error).”

This means that the results should be interpreted as “richer and bigger” households have a stringer electricity response to heat and a smaller water response to heat. The water results are still really interesting in this case. It means that smaller households shower disproportionately more than bigger and richer households. The electricity heterogeneity is a little bit less interesting through this, I think. Or convince me of the opposite. Bigger apartments with more people in them physically require more electricity to get cooling services per capita because of the increased volume of air that needs to be cooled. I am seeking a much stronger explanation of the impact of the “per capita” versus “higher income” effect here.

2. I would like to also suggest thinking about the interpretation of the results. While I believe the empirical findings of the paper - namely that poor people increase their water consumption in response to heat events, I wonder if this is in some sense a “convergence” response. If rich people find water relatively inexpensive and “shower at will” while poor people are more hesitant due to the cost, maybe both have the same “water consumption bliss point”. Maybe two showers a day is the bliss point when it’s hot. Poor people do not consume that much but rich people do. What this means is that as incomes rise, poor people consume water up to that bliss point and then install air conditioners and in addition to showering more (up to the bliss point), consume air conditioning services. This suggests a consumption ladder in some sense, which is similar to what the appliance adoption literature has observed. Table S1 in the supplemental materials sort of gets at this. I am not sure you can test for this, as you do not know how many people live in the bigger apartments and hence income is confounded by number of occupants. But it would be important to talk about what this means in the larger economics/income literature. Can we expect water consumption to plateau and electricity consumption to grow nonlinearly (in temperature)? In the introduction you talk about them almost as if they were substitutes, which I am not sure is the only/right interpretation.

These are thoughtful points. I am very grateful to this reviewer, as well as the two other reviewers, for pushing me to strengthen the Discussion section. Here I reproduce statements that I added on p.12 that speak to “convergence” and to “per capita versus higher income” (assuming my understanding of the reviewer’s request is correct):

“As incomes and AC adoption increase across other apartment types, the +1 °C-induced shift in water demand, while positive, declines in magnitude: (i) +5.3 L/d for 2-room apartments; (ii) +4.8 L/d for 3-room; (iii) +4.3 L/d for 4-room; (iv) +2.7 L/d for 5-room; and (v) +0.6 L/d and statistically insignificant for condominium apartments. Since occupancy tends to grow from 1-room to 5-room apartments, the drop in the magnitude of the *per capita* water demand increase is even steeper. One interpretation of this result is that energy-intensive AC can substitute for water-based cooling services and there are systematic, predictable differences in income and AC adoption (on average) across the different apartment types (Figs. 5B and S.10). An alternative

interpretation would be that of a convergence response, with high-income individuals “showering at will”—say two daily showers irrespective of heat—whereas cost-sensitive low-income individuals would raise their water use—to a “bliss point” of two daily showers, say—only during hot weather. Both interpretations are consistent with the evidence presented in Fig. 5A and Table 2 that population subgroups with lower income, lower (housing) wealth and/or lower AC access increase their water consumption in response to routine heat. While testing between these and other alternative interpretations is beyond the scope of this research, the pattern documented in Table S.2—that indoor water use after adjusting for differences in mean household size on average declines from 1-room to condominium apartments—does not seem to favor the convergence hypothesis.”

3. *How does your temperature response for wealthy household’s electricity consumption compare to what others have found in the North American and European context? The response should be in the same ballpark. It would add credibility to the study to point this out.*

On p.13, I added:

“This (condominium apartment) subgroup’s high income and AC saturation may explain why the temperature response for this climate zone is about double that typically estimated for a US population^{REF}. To add further context, a study of Singapore schoolchildren equipped with portable sensors estimated that median (resp., mean) AC exposure over a 24-hour cycle is 4 hours (resp., 6 hours)^{REF}.”

References added (beyond references previously included elsewhere in the manuscript):

Sailor DJ, Pavlova AA. Air Conditioning Market Saturation and Long-Term Response of Residential Cooling Energy Demand to Climate Change. *Energy* 28, 941-951 (2003).
Happle G, Wilhelm E, Fonseca JA, Schlueter A. Determining Air-conditioning Usage Patterns in Singapore from Distributed, Portable Sensors. *Energy Procedia* 122, 313-318 (2017).

4. *This is the first paper that actually gets at the water - energy nexus in a way that is actually interesting in my view. I would think about pitching this around that mystical concept. At least in the introduction.*

In response to this valuable suggestion, on p.14 I added:

“The US Department of Energy has called for increased integration in the analysis and development of interconnected water and energy systems—the so-called “water energy nexus”—most notably from an infrastructure perspective including the energy consumed in supplying water^{REF}. The agency argues that climate change, including shifting temperature extremes and rainfall variability, strengthens the case for a coupled approach to regulating use of strained water and energy resources including associated greenhouse gas emissions. One can view the present research as taking the water-energy nexus to the household domain, particularly in the context of an urbanizing tropical landscape undergoing both climatic and human transitions. Documenting a water-energy nexus in household heat relief amenities, which varies by socioeconomic group, is the article’s key contribution. It adds to a nascent literature examining temperature dependent consumption patterns focused largely on North America and Europe^{REF}.”

Reference added:

DOE. *The Water-Energy Nexus: Challenges and Opportunities*. U.S. Department of Energy (2014).

Heal G, Park J. Temperature Stress and the Direct Impact of Climate Change: A Review of an Emerging Literature. *Review of Environmental Economics and Policy* 10, 347-362 (2016).

5. In the first paragraph I would make it clearer that there is a quantity and quality issue arising from climate change - both for water and electricity (if we think of reliability as a quality issue).

Thank you for encouraging me to improve the opening paragraph. However, it is not clear to me what angle to the quantity and quality/reliability issue you would advise me to highlight. For now, I added the following sentence, and would be happy to consider a different angle/set of references that you may have in mind:

“Climate change may impact both the quantity of resources—for example, air conditioning demand—and the quality of resources, such as the reliability of supply, including electricity^{REF}”

References added (beyond references previously included elsewhere in the manuscript):

Auffhammer M, Baylis P, Hausman CH. Climate Change is Projected to have Severe Impacts on the Frequency and Intensity of Peak Electricity Demand across the United States. *Proceedings of the National Academy of Sciences* 114, 1886-1891 (2017).

Mideksa TK, Kallbekken S. The Impact of Climate Change on the Electricity Market: A Review. *Energ Policy* 38, 3579-3585 (2010).

6. On your page 2, you claim that showers and washing machines are more widespread than ACs. You should cite something from the appliance adoption literature that shows this to be true.

I clarify that the statement relates to Singapore’s households. Showerheads are present in almost every home; bath tubs are uncommon. (This was confirmed in the 300-person survey I implemented in response to Reviewer #1 general point.) Fig. 1A, referenced in the same paragraph and based on the quinquennial Household Expenditure Survey (2012/13 HES), documents that “(a)t a given expenditure level, washing machines enjoy more widespread diffusion than AC.” In response to this point, I added “(Fig. 1A)” to the end of this sentence. I also reference a study by the water agency, based on installing faucet/appliance-level meters over several weeks at a sample of households across different dwelling types, which quantified the importance of ablution to Singapore’s home water use:

“An observational study by the water agency of end uses in 2016/17 found that showering accounts for 27% of Singapore’s home water use, followed by bathroom tap/basin (18%), flushing (18%), kitchen (16%) and laundry (15%).^{REF}”

Reference added:

PUB. *Showering Tops Water Usage in Household Water Consumption: Singapore Household Water Consumption Study 2016/2017*. Public Utilities Board, Singapore’s National Water Agency (2018).

7. In maybe a follow up project, can you identify movers and use those as a source of identification? If not, you should make that clear in the manuscript.

Thank you for the suggestion. In such a study, one can perhaps focus on a narrow window around the move where income is less likely to have shifted, as this might otherwise co-explain shifts in appliance adoption and resource use. Reviewer #1 also requests more clarity on this point. I added the following sentences on p.3:

“Thus, for clarity, each account number corresponds to a unique combination of user and dwelling. By design (account number), the data controls for—but does not track—a moving family over different dwellings.”

8. *Page 5 reports that mean household...*

This comment may have been cut off. In response to a comment by Reviewer #3, the discussion now on p.4 of mean household income per capita and AC penetration, as well as mean household size, by apartment type is clearer about the source of this data, the quinquennial Household Expenditure Survey. This had previously been stated only in the reference and in the figure/table captions. I also try to be clearer on what sample (by apartment type) the mean is taken over:

“The 2012/13 HES^{REF} reports mean household income per capita and overall AC penetration by apartment type. Based on this source, Table S.2 (and Fig. 5B below) report that both mean income per capita and AC adoption...”

9. *I was puzzled by the m^3/kWh measure, but really like it now!*

Thank you for pointing this out. In response to Reviewer #3’s request to be clear about the different data sources, the text describing the m^3/kWh pattern by apartment type now reads:

“Table S.1 describes a water-to-electricity quantity ratio in the microdata—in arbitrary m^3/kWh and averaged across household by period observations within apartment type—that decreases monotonically over the different apartment types (i) to (vi), from 0.113 for 1-room apartments compared to a lower 0.033 for condominium apartments. This pattern is in agreement with the water and electricity burdens reported in the 2012/13 HES (Fig. 1B).”

10. *Stick with a single currency in the paper.*

I have changed all currency units to US\$ (except momentarily when explaining the income bins in the Household Interview Travel Survey).

11. *Why is the last day of the bill the right assignment of month for your measure of seasonality?*

I start by noting that there is significantly less seasonal variation in tropical Singapore compared to a temperate climate such as Europe or North America (Fig. 2). To p.17, I added:

“In terms of climate, it is worth noting that there is significantly less seasonal variation in the tropics compared to a temperate climate. While this feature limits the range of temperature over which households respond—for example, we do not learn about impacts of winter temperatures in Europe—it provides a “natural” control for the key climate component of seasonal variation.”

In response to this comment, I report below on estimates for specifications as in Fig. 5A (the main result) except that “**month** fixed effects, based on the **last day of period t** ” are replaced by:

- (i) “**week** fixed effects, based on the last day of period t ”—these more granular controls soak up variation in temperature, the key variable of interest, but the findings are robust.

(ii) “month fixed effects, based on the **midpoint of period t** ”

To p.14, I added:

“Estimates are similar if month fixed effects are based on the midpoint of period t rather than its last day, and if I specify (52) week fixed effects rather than (12) month fixed effects.”

12. Why is clustering by household right? Your unit of treatment is billing period and weather station? Justify your standard errors.

Singapore being an island with an area of 720 km², there is limited weather variation. The variation I exploit—for different household (apartment) types—is within-household temporal co-variation in environmental conditions and resource demands over billing cycles, with billing cycles starting and ending on different dates across different households. I explain this on p.15:

“For example, for a given household i , electricity billed in March, April and May 2015 was based, respectively, on actual (A, 350 kWh), estimated (E, 370 kWh) and actual (A, 390 kWh) meter readings on March 5, April 5 and May 5. I then observe *actual* use of $370 + 390 = 760$ kWh over the 61-day period t that starts on March 6, 2015 and ends on May 5, 2015. Electricity use is $760/61 \times 30 \approx 374$ kWh per 30 days. The estimation sample (for electricity) would include one observation of 374 kWh/month for the bimonthly period March 6 to May 5, 2015. Since SP Services staff distribute their visits evenly across residences, usage periods vary across households, for example, the 61 days closing on May 1 for some households, the 61 days up to May 2 for others, and so forth (Fig. S.2). This adds to the sample variation in heat exposure^{REF}.”

In response to this comment:

(i) A new Fig. S.4 reports the distributions of the number of periods of water use and electricity use observations across households. For brevity, I do not reproduce this here.

(ii) On p.1 I better describe the geographic context and structure of the microdata:

“Here I examine several years of microdata on water and electricity bills for a one-in-ten random sample of households in Singapore, namely 120,000 households with a modal 19 bi-monthly periods of observation per household (microdata for short). Singapore (population 5.4 million) is a leading, newly affluent island city-state covering 720 km² off Peninsular Malaysia in tropical Asia.”

(iii) I justify the standard errors on p.7:

“Standard errors, reported in parentheses under point estimates, are clustered by household. This allows unobserved demand determinants to correlate over time within household, with 19 periods being the modal number of observations per household (Fig. S.4).”

(iv) The figure below repeats the specification reported in Fig. 5A except for adopting alternative standard errors based on two-way clustering on household and usage period, i.e., the same first day besides same last day of the period, such as March 6 to May 5, 2015. Confidence intervals are more conservative, but the finding is robust.

Repeat Fig. 5A, change in water use against change in electricity use, by apartment type, with alternative (more conservative) 95% CI based on two-way clustering on household and usage period.

13. The table and figure notes are super excessive. The tables are also too big. I would edit these down and move what I could to the SI.

Thank you for this suggestion, which is related to a suggestion by Reviewer #3. I cut down on table and figure notes, as well as on table content, moving what I could to the Methods section (now included at the end of the manuscript, per journal guidelines) or to the Supplement. I moved one-third and one-half of the regressions in Tables 1 and 2, respectively, to Supplementary Tables S.4 and S.6.

In this response letter to Reviewer #3, **verbatim edits and additions to the manuscript are highlighted in yellow**. I omit minor edits intended to improve readability and that do not change content.

Reviewers' comments (in italics):

Reviewer #3 (Remarks to the Author):

The paper analyses the dependence of water and electricity demand on climatic factors, keeping into account the income (included as a proxy, through information on the apartment-type).

It is a very interesting subject, extremely novel since not yet explored, to my knowledge in the literature and of extreme importance in terms of both real-time forecasting of water and energy consumptions (for the multi-utilities) and of future scenarios of customers' needs, for policy-makers.

I am grateful for these words of encouragement.

On the other hand, the paper is pretty repetitive: from the abstract on, the key result (that the increase in temperature makes the rich households to increase air-conditioning and makes the poorer families increase the water use) is repeated too many times.

Thank you for raising this. I have reviewed the text to cut back on the repetition, beginning with your suggestion below (and that of Reviewer #1) to omit the paragraph on results from the introduction.

In addition, the results are probably more specific than claimed, since the situation of Singapore, in terms of the combination climate + income, is probably pretty unique.

This is a fair point. Besides increasing (somewhat) the labeling of climate as tropical throughout the manuscript, in the Discussion section I added the following caveat (also borrowing from the reviewer's opening remarks on real-time forecasting), before describing how Singapore leads other cities in tropical Asia:

“Further work should investigate whether the empirical findings for Singapore extend to other urban populations in differing climates and/or levels of development, helping improve real-time demand forecasting for multi-utilities. For example, other cities in tropical Asia experience similar climates but lag Singapore in terms of economic development^{REF}.”

In the Introduction, I now reference the world's large tropical population, which has yet to adopt AC:

“Currently, only 8% of the world's tropical population, three billion strong, has AC^{REF}.”

Reference added:

Economist. Air-conditioners do great good, but at a high environmental cost. *The Economist* August 25, (2018).

Yuen B, Kong L. Climate Change and Urban Planning in Southeast Asia. *Surveys and Perspectives Integrating Environment & Society* 2, (2009).

Another issue that should be clarified is why the author assumes (at least initially since eventually the results do not conform it) that there should be an influence between air pollution and water use: I confess that the link is not apparent to me...

Thank you for sharing this. I added statements to this effect on p.5 and in a subsection in Methods, titled “Inclusion and plausible endogeneity of PM2.5,” now at the end of the manuscript (a shorter version was previously included in the Supplement):

“Residential demand for water and electricity may shift with air pollution through avoidance behavior, for example, high (and visible) particle concentrations may induce households to stay at home or close the windows, thus flushing the toilet and using the AC more often.”

“Singapore experiences variable levels of air pollution from both onshore and transboundary sources, including very high PM2.5 in mid 2013 and late 2015 due to land fires in equatorial Asia (Fig. 2D), and influenced by atmospheric ventilation conditions. Households’ avoidance behavior implies that pollution is a driver of household utility demand^{REF}. For example, in response to poor air quality, households may stay more at home, flushing the toilet and eating meals at home more often, or may shut the windows, turning on the AC. Thus, PM2.5 is a control in the household resource demand equation [1].”

Related to this point, on p.7 in the Results section I briefly describe other environmental determinants of water demand:

“The lower temperature effect in specifications 4 and 6 relative to specification 3 illustrate the importance of controlling for PM2.5 when examining the weather determinants of residential resource use. This is an important point that merits highlighting. Moreover, I find that a 10 $\mu\text{g}/\text{m}^3$ increase in average PM2.5 during a usage period increases water demand by 0.5% (omitted from Table 1 for brevity). Controlling for the incidence of rain, conditions that are more humid—higher relative humidity, lower dew point depression—raise water demand.”

Most importantly, the paper should better describe the available data sets (this is crucial when applying any data-driven methodology) and also the applied statistical techniques, in order to better justify the conclusions that are drawn. The information on the data is scattered in the different sections in a not orderly way, often AFTER having presented some figures based on them or some results.

These are good suggestions, which I incorporated (and respond to point by point below).

And both in the Table description at p. 8 and in Section V: the discussion of the results are pretty heavy to follow, with too many numbers reported in the text: a few illustrative figures should make the paper much shorter and easier to follow (after all, the scientific content, even if interesting, is not so much...). I would suggest to summarize the results into a much shorter technical note, with a clearer description of the data and the methodology and a more visual, friendly description of the results.

I understand and respect this view. Manuscripts written for mainly economics audiences are usually heavy on their use of tables, heavy on in-text references to tables and numbers, and light on figures. But I am not targeting a mainly economics audience.

So here is what I did in response to this feedback. I moved the Methods section to the end of the manuscript, complying with the journal’s style guidelines. The Results section now flows continuously from the section titled “Institutions and data combined from different sources,” without requiring the reading of Methods.

I shortened the two tables of estimates, moving one-third and one-half of the regressions reported in the original Tables 1 and 2 to the Supplement, now in Tables S.4 and S.6. I now report the original Table 1 results by way of two new figures, Figs. 3 and 4. All figures show 95% confidence intervals of percent increases in resource use (lighter to read) rather than point estimates

and standard errors in log points (heavier). The text also discusses percent increases, not log point changes, in resource use.

I reduced the numbers reported in the text. For example, I focus on °C units for temperature rather than additionally providing the °F equivalent. I added more intuition in the discussion of results. I reduced the use of acronyms for technical terms, writing “fixed effects” rather than “FE.”

A further option, which I defer to the editor, is to label the tables of results as “expanded tables” rather than (main) “tables.” My understanding is that *Nature Communications* requests that all results be included in the manuscript, not in the Supplement.

DETAILED COMMENTS

p. 2, l. 8: do not refer to “log-points” but to the values in their actual unit and the percentage of increase/decrease.

The discussion of the results now refers to percent increases.

p. 2, ll. 5-11: there is too much detail for an introduction section, too much of the results is anticipated in this paragraph in quantitative terms, that should instead go only into the Discussion of the results section.

Thank you for this suggestion, also made by Reviewer #1, which I now follow. The original paragraph in the introduction followed the convention in the economics literature of describing one’s results after discussing the context and literature.

p. 2 (beginning of Section 2) and Fig. 1A: Please explain which data were used for obtaining the numbers shown the figure? Expenditure data are not provided by the utility of course.

I apologize for not being clear. I now state:

“Fig. 1A is based on Singapore Department of Statistics’ quinquennial Household Expenditure Survey, 2012/13 HES for short^{REF}. The plot shows that the adoption of electricity and water consuming appliances...”

and, in the subsequent paragraph,

“Also based on the 2012/13 HES, Fig. 1B shows that that the electricity share of household expenditure...”

p. 3, l. 3-5: Is water tariff dependent on the monthly volumes? This is partly explained at p. 4, but it should be here.

Thank you for pointing this out (Reviewer #1 makes a similar point). In response, when first referencing Fig. 1C on p.3, I added the statement:

“Residential electricity and water prices, described below and reported over time in Fig. 1C, are essentially constant over quantity and hour of use, and are invariant across households.”

p. 3: explain here (and not at p. 4 & 5) which other information is available on the costumers (apartment size and value, number of members in the family,...).

I moved the information that is contained in the microdata, regarding apartment types, as well as two-digit zip code, to p.4. I similarly moved the paragraph on limitations of this dataset to p.4, starting with:

“The microdata does not inform on the individual household’s income, size or its adoption of appliances.”

I am now more specific throughout when referring to the different sources of data, in particular the microdata = household-level panel (water and electricity use by household over time, time-invariant apartment type), as distinct from the 2012/13 Household Expenditure Survey (mean household income per capita, mean household size and overall AC penetration by apartment type). On p.1 I label the study main dataset as follows:

“Here I examine several years of microdata on water and electricity bills for a one-in-ten random sample of households in Singapore, namely 120,000 households with a modal 19 bi-monthly periods of observation per household (microdata for short).”

Finally, my response to Reviewer #1 “Page5 / Para3” that the microdata does not inform on a household’s income may be relevant to this point. For brevity, I do not reproduce the response here.

p. 4: there is information on water conservation measures and tariffs but not on energy conservation tariffs and energy tariffs.

Thank you for your careful reading. This is partly related in part to your point above. The statements describing resource conservation efforts and tariffs now cover both water and electricity, namely:

“There has been no discernible variation in household resource conservation measures or public information campaigns...”

“All households face the same marginal price schedule, both for water and for electricity. Marginal prices are either constant (electricity) or essentially constant (water) over quantity supplied.”

p. 5: Table S.2 and Fig. 3: where does this information come from? The source is not clear. In addition, the number of members of the family is a very important information too and it seems to be missing...

This point is related to points made above. I have expanded the text as follows:

“The 2012/13 HES^{REF} reports mean household income per capita and overall AC penetration by apartment type. Based on this source, Table S.2 (and Fig. 5B below) report that both mean income per capita and AC adoption...”

p. 5, l. 10: actually how do we know if the apartments are naturally ventilated or not? The assumption of the relationship between apartment size/income/no air conditioning has not been proved or sustained by actual data...

From the 2012/13 Household Expenditure Survey, we learn that only 14% of households in 1- or 2-room apartments have air conditioning. The changes described above should make this clear. Further, I replaced “naturally ventilated 1-room apartments” by “mostly AC-free 1-room apartments.”

With regard to ascertaining a household’s socioeconomic standing—broadly defined to encompass wealth and expenditure in addition to income—for brevity I refer the reviewer to my response to Reviewer #1, “Page5/Para3” on why a household’s dwelling type is a good proxy for socioeconomic group, a construct that is now expanded in the text on p.2 and p.4.

p. 5: the information in terms of percentages is not fully informative: there is the need to provide the results terms of both volumes and percentages (and stating that 25% of the small (‘poor’) households use 20% of water volumes does not support the conclusion that poor family consume more water... Data reflecting both actual volumes and number of family members are needed.)

The discussion is offered in a “difference in difference” (and “free of family size”) sense: small apartments’ water versus electricity shares of aggregate use relative to big/more premium apartments’ water versus electricity shares. Related to this discussion, Reviewer #2 wrote: “*I was puzzled by the m^3/kWh measure, but really like it now!*” Thus I added the highlighted statements to the text in the hope of making it more readily understandable, as well as bringing in statistics on average occupancy (previously reported only in Table S.2):

“The microdata and 2012/13 HES, summarized in Tables S.1 and S.2 respectively, jointly indicate that average electricity use in lower-income, mostly AC-free 1-room apartments is 21% that of high-income, AC-saturated condominium apartments (134 against 622 kWh/month). In comparison, water use in 1-room apartments is 59% that of condominium apartments (9.8 against 16.5 m^3 /month). The 2013 HES further indicates that average occupancy in 1- or 2-room apartments is 62% that of condominium apartments (2.1 against 3.4 persons).”

Apartments with up to three rooms make up 25% of the random sample’s observations, yet account for 20% of aggregate water consumption vs. 15% of aggregate electricity consumption. At the high end of the income distribution, the condominium apartments that comprise 17% of observations account for 16% of aggregate water consumption vs. 25% of aggregate electricity consumption. This pattern is informative in a “difference in difference” sense: smaller apartments’ water versus electricity shares of aggregate use relative to larger/more premium apartments’ water versus electricity shares. The microdata indicate that electricity use increases more steeply over the socioeconomic distribution than does water use.”

p. 5, l 22: DOS? Please put the source in clear and adding the year of the publication. Clarify also which information is available on family size? A mean depending on apartment size?

On the latter question, yes. All clarifications requested have been added to the text, per the responses above.

p. 5 (end): Can you discuss the differences between the results obtained in the present case study and in the Santa Barbara one? (and note that for the case study of Santa Barbara the actual volumes - and not percentages - are correctly reported. ...)

In response to Reviewer #1 (Page5 / Para4), on p.7 I now describe residential water quantities for more “comparative examples”—Korea in 1982 and Adelaide City in 2000—than the OECD and Santa Barbara studies cited originally.

I kept the Santa Barbara comparison as it collected panel data and many variables (albeit on a small sample of high-density houses) and: (i) it did not find an association between household income and water demand, and (ii) despite being able to collect many household variables, it did not survey electrical appliances or electricity use.

Section II is not clear, more details are needed on the statistical procedures.

Eq. 1: All the terms presented in equation 1 should be defined either before or just following the equation and not too much later on.

Text following Eq 1:

- The phrase on the meter readings is not clear: how do you go from bimonthly to monthly data: assuming the same values for the two months?

- Not clear what $T(t)\lambda(i)$ means: isn't the apartment typology represented by λ alone?*

p. 7, End of Section 3: please define what “2SLS” means.

p. 8: “interact the variable”?? what does it mean?

Thank you for these suggestions that I provide more clarity on the methods used. Following the journal’s guidelines, I moved the Methods section to after the Discussion. This allows for more space. I brought in two sections on methods, titled “Inclusion and plausible endogeneity of PM2.5” and “Linearity of temperature effects,” that were previously in the Supplement. I also edited the text in Results and Discussion to reflect that at that point the reader may not yet have read Methods. I followed your suggestions as follows:

(i) All the terms presented in equation 1 are now defined just following the equation (and elaborated later). For convenience, I reproduce the text that follows equation 1 (p.15):

“An observation is an individual household i by period t of usage observed between two actual meter readings. This estimating equation relates the logarithm of utility use, $water_{it}$ (or $electricity_{it}$), to observed drivers, $(T_t, X_{it}, \phi_i, \delta_t)$, as well as unobserved determinants, ε_{it} , of demand. Observed demand determinants, further described below, include: ambient temperature T_t , that enters the relationship according to a potentially nonlinear and household-specific function $f_i(\cdot)$; other potential environmental drivers X_{it} ; time-invariant shifters that differ by household ϕ_i ; and time-varying variables δ_t . (λ_i, α_i) are parameters that govern the relationship between environmental conditions and utility demand.”

(ii) I expanded the text that explains how I handle the approximately one-half of bills in the data that are based on estimates: I simply integrate over them. Observations in the water and electricity estimation samples have a modal duration of 61 days. I do not “produce” two monthly observations from a bi-monthly period of actual use, since I do not separately observe actual use over the first month and the second month. (This would be akin to “producing” daily observations from data observed at monthly resolution.) I reproduce the expanded text on p.15, which includes an example and a new figure:

“With regard to household by period units of observation, the vast majority of periods have a bi-monthly duration. SP Services bills customers every month, but only about one-half of these bills are based on actual readings, which necessitate (per the current processes and technology deployed) that an SP Services staff physically visit the dwelling to read water and electricity meters. These meter-reading visits to dwellings are undertaken once every two months, and the interim monthly bill is based on estimated use^{REF}. The microdata includes all monthly bills to each sampled household, further stating which bills are based on actual readings and which are based on estimates. Since my purpose is to examine actual use, I integrate billed use over each period elapsed from one observed actual reading to the next. I express use as a 30-day rate, i.e., in terms of a monthly equivalent in m^3 /month or kWh/month.

For example, for a given household i , electricity billed in March, April and May 2015 was based, respectively, on actual (A, 350 kWh), estimated (E, 370 kWh) and actual (A, 390 kWh) meter readings on March 5, April 5 and May 5. I then observe *actual* use of $370 + 390 = 760$ kWh over the 61-day period t that starts on March 6, 2015 and ends on May 5, 2015. Electricity use is $760/61 \times 30 \approx 374$ kWh per 30 days. The estimation sample (for electricity) would include one observation of 374 kWh/month for the bimonthly period March 6 to May 5, 2015. Since SP

Services distribute their visits evenly across residences, usage periods vary across households, for example, the 61 days closing on May 1 for some households, the 61 days up to May 2 for others, and so forth (Fig. S.2), adding to the sample variation in heat exposure^{REF}. Email correspondence with SP Services informed that longer periods than two months happen due to “a utility meter (being) inside the customer’s premises where (the) meter reader cannot access (when visiting), ... (or the) meter is faulty. In such cases, investigation will take place and the faulty meter will be replaced. But until the replacement is complete, the readings can only be estimated.” In such cases, I integrate over several monthly bills to the next meter reading. Shorter periods than two months are due to (relatively few) customers calling in with actual readings within one month of a meter reader’s visit, in time for the next bill. Fig. S.3 shows the distributions of duration for water and electricity (actual) use observations.”

(iii) I expanded the text explaining the term $T_t\lambda_i$ and what I mean by interacting a variable. Thank you for requesting greater clarity. For convenience, I reproduce the edited text (p.16) here:

“I allow the impact of temperature on utility use to vary by apartment type, thus interacting T_t (or a nonlinear function of T_t) with a set of indicators for the different apartment types. In a linear specification, justified by the narrow tropical temperature range, an apartment type-specific temperature effect is denoted $T_t\lambda_i$, i.e., $T_t\lambda_{1-room}$ if the household lives in a 1-room apartment, $T_t\lambda_{2-room}$ in a 2-room apartment, and so on. A linear specification would not be applicable if one were inferring responses to North American winter and summer temperatures in the same regression model.”

(iv) Relatedly, I explain the meaning of “interact” on p.8 (Results) by adding the highlighted text: “I interact the variable of interest, average temperature during the usage period, with a set of indicators for the different apartment types, as proxies for different socioeconomic groups. This allows estimated heat impacts to differ by group.”

(v) I now refer to two-stage least squares (2SLS) and ordinary least squares (OLS) on first mention, both in the Methods section (after Discussion) and in the Results section. Relatedly, I provide more intuition, in my specific context, on the 2SLS estimator, on p.18:

“A concern that may arise when adding PM2.5 as a regressor is “reverse causality”: electricity or water demand shocks ε , unobserved by the empirical researcher, may lead to higher emissions and pollution, e.g., from electricity generation to meet electricity demand, or economic activity that shifts both resource demand and PM2.5 via emissions. On top of this, PM2.5 may correlate with temperature, the main variable of interest. In addition to OLS, which restricts regressors to be orthogonal (conditionally exogenous) to unobserved demand shocks, I present results based on an alternative estimator that allows PM2.5 to be endogenous. The two-stage least squares (2SLS) estimator I specify assumes that a household’s resource use responds to shifts in regional land fires and shifts in atmospheric ventilation only *indirectly*, through these variables’ impact on ambient PM2.5. These “instrumental variables” are assumed to be orthogonal to unobserved demand shocks. Intuitively, co-variation between household demand and the instruments works through pollution to demand, producing unbiased estimates of the impact of pollution and its correlates such as temperature.”

(vi) To Fig. S.9, now labeled “allowing for endogenous PM2.5 (and temperature)...,” I added a further robustness test to allow for attenuation bias from possible classical measurement

error in the main variable of interest, temperature. I instrument for temperature as measured by Meteorological Service Singapore with temperature as measured by an independent source, the National University of Singapore’s Department of Geography, at its Kent Ridge campus weather station. I explain this robustness test in the expanded Methods section on p.19:

“Testing for measurement error in temperature. To check for possible attenuation bias from “classical measurement error” in the main variable of interest—temperature, as measured by Meteorological Service Singapore (MSS)—I perform the following robustness test. I add T_t^{MSS} (temperature measured by MSS) to the vector of endogenous variables, and add temperature as measured by an independent source, T_t^{NUSG} , to the vector of instruments. The superscript denotes the Kent Ridge weather station maintained by the National University of Singapore’s Department of Geography. While NUSG’s site on the vegetated and less urban Kent Ridge campus is slightly cooler than the average MSS weather station (Table S.3), the independently measured temperature series are very tightly correlated, suggesting that attenuation bias is not present, as indeed I find.”

Attention Check

We care about the quality of our survey data and hope to receive the most accurate measures of your opinions, so it is important to us that you thoughtfully provide your best answer to each question in the survey.

Do you commit to providing your thoughtful and honest answers to the questions in this survey?

- I can't promise either way
- I will not provide my best answers
- I will provide my best answers

Hypothesis

These page timer metrics will not be displayed to the recipient.

First Click: *0 seconds*

Last Click: *0 seconds*

Page Submit: *0 seconds*

Click Count: *0 clicks*

It is a **very hot weekday** in Singapore this September.

Your **daily routine has been normal**, attending school or work, running errands, going to the community centre, and so on.

Which of the following may **apply to your home that day or evening**? Please tick at most **three** (one, two or three) statements that are more likely to apply:

- Due to the heat, at home I **washed my face more often or for longer**, compared to a cooler day.

- Due to the heat, at home we/I **showered more often or for longer**, compared to a cooler day.
- Due to the heat, at home I **put wet towels in the refrigerator to place on my body once cool**, compared to a cooler day.
- Due to the heat, I **spent more time at home**, compared to a cooler day.
- Due to the heat, at home we/I **put more dirty/sweaty clothes in the washing machine**, compared to a cooler day.
- Due to the heat, at home we/I **turned on the electric fan**, compared to a cooler day.
- Due to the heat, at home we/I **shut the windows and turned on the air conditioner**, compared to a cooler day.
- Other
- None of the statements above are more likely to apply at home due to the heat, compared to a cooler day.

Demographics

Tell us about yourself and your home:

Age:

Gender

- Male
- Female
- Other

Position in the household (select **one**):

- Head of household
- Spouse of head of household
- Son/daughter of head of household
- Parent of head of household
- Flat-mate / room-mate of household

Type of dwelling / premises I live in (select **one**):

- HDB Flat 1 Room or 2 Rooms (1 bedroom)
- HDB Flat 3 Rooms (2 bedrooms + Living/ dining area)
- HDB Flat 4 Rooms (3 bedrooms + Living/ dining area)
- HDB Flat 5 or 6 Rooms, Executive Flat or Multi-generation flat
- Condo (Condominium apartment)
- Landed property (Townhouse, terrace or semi-detached house, bungalow)
- I don't know

Appliances at home (**tick all that apply**):

- Air conditioner
- Electric fan
- Shower
- Washing machine
- Bath tub

Powered by Qualtrics

Reviewers' comments:

Reviewer #1 (Remarks to the Author):

The author has made substantial revisions to the manuscript from its initial submission.

These revisions reflect a number of minor grammatical and syntactic revisions to the text that were suggested by the reviewers, as well as several other more substantive revisions to the methodology description, discussion, and cited references.

This reviewer's primary concerns were addressed through the inclusion of new text:

- citing the possibility of alternative personal cooling strategies within the home
- discussing the broader significance of integrated energy-water usage issues within the Singapore's specific context and resource challenges

Based upon the content of these various revisions I would recommend that the manuscript is ready for publication.

Reviewer #2 (Remarks to the Author):

I am very happy with the revisions and only have two minor editorial comments, which you can feel free to take or leave. This is a very nice paper!

1. I think the sentence you added in response to another reviewer "A study of 653 rural households in Iran collected information on water consumption and AC adoption but, while mentioning a possible interaction, does not relate the two variables²⁴." is not necessary and breaks the flow.

2. The sentence (Water and electricity expenditure shares that decline in income are noted—though not compared—for US households in Timmins 33.) also interrupts the flow.

I really appreciated the streaming of the figures and tables. The paper is much more readable. Also the discussion is much better.

Reviewer #4 (Remarks to the Author):

I was asked not to review the paper freshly, but to restrict myself to addressing whether the authors' revisions adequately addressed reviewer #3's comments and concerns have been adequately addressed.

I feel that the revised manuscript has adequately addressed most of these concerns. However, there are two areas where I think a little additional work could better address the concerns.

1. In justifying the linear relationship posited between temperature and utility use, the author relies on the comparatively narrow range of temperature variation in tropical climates (as opposed to the greater range typical temperature climates). This is plausible but not necessarily true. It would be better to consider goodness of fit (either with a formal quantitative goodness of fit test or a qualitative examination of residuals versus predicted utility use).

2. While the author addressed specific comments by the reviewer about the general lack of clarity in the language, I still found that the reviewer's concerns that the language is often stilted in a way that makes it difficult to read and to understand exactly what the author did. One example: on page 12, the author refers to a "300-person survey," which is described only in the caption to a table in the supporting information. It would be helpful to include a short description in the main text of what the survey was (especially since the article draws on many different surveys and other data sources so the reader may have trouble keeping them all straight), and there should be a textual description of the survey in the supporting information, not just a table caption.

Elsewhere, the phrasing of "I interact the variable of interest with ..." remains. The author added explanatory material, but the English is still awkward and would be better phrased along the lines of "I considered interactions between the variable of interest and ..." There are many places throughout the article where it could benefit greatly from the attention of a good copy editor. These

issues are not severe enough to hurt the scientific validity of the paper, but they are likely to diminish the impact of the paper by putting off readers who might otherwise find it interesting.

3. On page 2, the author writes, "The evidence suggests that hot weather induces individuals to wash more ..." without a citation. It is unclear whether the author is talking about the results of this paper (in which case this does not belong in the introduction) or about other research (in which case a citation would be required). This ties into the first and third reviewer's comments about overloading the introduction, which makes the paper repetitive.

Reviewers' comments (in italics):

Reviewer #1 (Remarks to the Author):

The author has made substantial revisions to the manuscript from its initial submission.

These revisions reflect a number of minor grammatical and syntactic revisions to the text that were suggested by the reviewers, as well as several other more substantive revisions to the methodology description, discussion, and cited references.

This reviewer's primary concerns were addressed through the inclusion of new text:

- citing the possibility of alternative personal cooling strategies within the home*
- discussing the broader significance of integrated energy-water usage issues within the Singapore's specific context and resource challenges*

Based upon the content of these various revisions I would recommend that the manuscript is ready for publication.

I am very grateful for Reviewer #1's suggestions and time throughout improving my work.

Reviewers' comments (in italics):

Reviewer #2 (Remarks to the Author):

I am very happy with the revisions and only have two minor editorial comments, which you can feel free to take or leave. This is a very nice paper!

1. I think the sentence you added in response to another reviewer "A study of 653 rural households in Iran collected information on water consumption and AC adoption but, while mentioning a possible interaction, does not relate the two variables²⁴." is not necessary and breaks the flow.

2. The sentence (Water and electricity expenditure shares that decline in income are noted—though not compared—for US households in Timmins 33.) also interrupts the flow.

I really appreciated the streaming of the figures and tables. The paper is much more readable. Also the discussion is much better.

I am very grateful for Reviewer #2's suggestions and time throughout improving my work.

With regard to the two "take or leave" minor comments on statements that affect the flow, on reflection I dropped the first statement—added in Round 1, thus contained in that round's response, to be published—but kept the second statement.

Reviewers' comments (in italics):

Reviewer #4 (Remarks to the Author):

I was asked not to review the paper freshly, but to restrict myself to addressing whether the authors' revisions adequately addressed reviewer #3's comments and concerns have been adequately addressed.

I feel that the revised manuscript has adequately addressed most of these concerns. However, there are two areas where I think a little additional work could better address the concerns.

1. In justifying the linear relationship posited between temperature and utility use, the author relies on the comparatively narrow range of temperature variation in tropical climates (as opposed to the greater range typical temperature climates). This is plausible but not necessarily true. It would be better to consider goodness of fit (either with a formal quantitative goodness of fit test or a qualitative examination of residuals versus predicted utility use).

In addition to the specification tests mentioned on pages 14-15 of my Round 1 response to Reviewer #1—who had requested further justification of the linear restriction—I now add quadratic specifications. Responses over the empirical support of average temperature variation, from 26 to 29 °C, are shown in a new figure, Fig. S.10, and referenced on p. 11, subsection “Robustness,” as follows:

“Further testing the linear-in-temperature restriction in Fig. 5A, I include (i) interactions between the square of temperature and apartment-type indicators (Fig. S.10, panel A), and, additionally, (ii) the square of each environmental control (Fig. S.10, panel B).”

Again for convenience, I reproduce the new figure’s caption here:

“Fig. S.10. Water vs. electricity demand responses to heat, by apartment type: Robustness to average temperature entering nonlinearly via a quadratic term. Departing from Fig. 5A’s specification, I progressively include: **A.**, Interactions between apartment-type indicators and the square of average temperature, and, additionally, **B.**, The square of each environmental control (average relative humidity, average dew point depression, average wind speed, average precipitation, and average PM2.5). The empirical support for average-period temperature is 26.1 to 29.1 °C, so I report 95% confidence intervals for the percent increase in water demand against the percent increase in electricity demand, by apartment type, for a 26.1 to 29.1 °C shift in average temperature.”

2. While the author addressed specific comments by the reviewer about the general lack of clarity in the language, I still found that the reviewer's concerns that the language is often stilted in a way that makes it difficult to read and to understand exactly what the author did. One example: on page 12, the author refers to a "300-person survey," which is described only in the caption to a table in the supporting information. It would be helpful to include a short description in the main text of what the survey was (especially since the article draws on many different surveys and other data sources so the reader may have trouble keeping them all straight), and there should be a textual description of the survey in the supporting information, not just a table caption.

Thank you for this suggestion. I added a description of the 300-person survey in the Methods section, on p.20. For brevity, I do not reproduce it here. An added benefit is to shorten the caption to Table S.7.

Elsewhere, the phrasing of "I interact the variable of interest with ..." remains. The author added explanatory material, but the English is still awkward and would be better phrased along the lines of "I considered interactions between the variable of interest and ..." There are many places throughout the article where it could benefit greatly from the attention of a good copy editor. These issues are not severe enough to hurt the scientific validity of the paper, but they are likely to diminish the impact of the paper by putting off readers who might otherwise find it interesting.

I reviewed all phrases using "interact" or "interaction," revising these along the lines suggested by the reviewer. I took the reviewer's advice and hired a copy editor (Barbara Nordin) to spot and improve awkward English. I do not list the changes here as they improve readability but do not change scientific content. Thank you for this suggestion.

3. On page 2, the author writes, "The evidence suggests that hot weather induces individuals to wash more ..." without a citation. It is unclear whether the author is talking about the results of this paper (in which case this does not belong in the introduction) or about other research (in which case a citation would be required). This ties into the first and third reviewer's comments about overloading the introduction, which makes the paper repetitive.

I thank Reviewer #4 for his/her careful reading and agree that this point ties in with the earlier advice regarding overloading the introduction with results. As such, I dropped the corresponding sentence.

REVIEWERS' COMMENTS:

Reviewer #4 (Remarks to the Author):

The revisions greatly improve the clarity of the manuscript and adequately address the comments and suggestions from my previous review. I recommend this manuscript for publication.